# Homoharringtonine exhibits senotherapeutic activity that mitigates diet- and age-associated obesity and insulin resistance and extends lifespan in mice

Eok-Cheon Kim[1,2,8], Han-Byul Jung [3,8], Yu-kyoung Park[1,3], Youlim Son [1,2], Hye-Na Cha[1,3], Yash Patel[4], Ju Hee Lee[4,5], Minah Choi[2], Soyoung Park [1,3], Il-Kug Kim[1,6], Lauren Pickel[4], Seungju Lee[7], Yuna Ha[7], Min-Gyeong Shin[3], Qiwei Zhang[4,5], Jielin Yang [4], Bruno Rodrigues de Oliveira[4], Nathaniel Vo [4,5], Annie Yew[4,5], Jacques Togo[4], Kafi N. Ealey[4], Su-Ryun Jung[1,3], Sunjin Moon[1], Hye-Jin Yoon[1,2], Jee-Young Lee[7], Hoon-Ki Sung [4,5] ✉, Jae-Ryong Kim [1,2] ✉ & So-Young Park[1,3] ✉

The accumulation of senescent cells in white adipose tissue (WAT) is closely associated with the functional decline of WAT and plays a causal role in the pathogenesis of metabolic diseases. Therefore, the elimination of senescent cells in WAT holds promise for the treatment and prevention of age-related metabolic diseases. Using a drug-repositioning strategy for 2150 clinically applied compounds, we discover that homoharringtonine (HHT), an FDA-approved anti-leukemic drug, manifests senotherapeutic activity in vitro in multiple cell types including human preadipocytes, while inflicting minimal cytotoxicity to non-senescent cells. HHT treatment prevents diet- or age-induced metabolic abnormalities in male mice targeting senescent adipocytes and preadipocytes to improve WAT function and reduce WAT inflammation. Moreover, HHT treatment attenuates age-associated phenotypes of human adipose tissue. Mechanistically, the senotherapeutic effects of HHT are mediated through the direct interaction of HHT with heat shock protein family A member 5 (HSPA5). Importantly, we found that HHT treatment delays aging and extends the lifespan in progeroid and aged mice. Our study demonstrates the novel senotherapeutic potential of HHT to mitigate age- and obesity-related metabolic dysfunction and extend longevity in mice.

Cellular senescence refers to a terminal state of growth arrest in which cells cease to divide, undergo morphological changes (enlarged and flattened shape), take on a senescence-associated secretory phenotype (SASP), and exhibit increased senescence-associated beta-galactosidase (SAβG) activity and expression of p53 and/or p16[1,2]. Senescent cells release inflammatory cytokines, growth factors, and matrix metalloproteinases, resulting in the dysfunction of neighboring cells[3,4]. Consequently, the accumulation of senescent cells induces tissue aging and organ dysfunction, thereby contributing to the development of age-related chronic diseases. Therefore, the elimination of senescent cells or the blocking of the SASP extends the healthspan and prevents the emergence of age-related pathologies[5–10].

The white adipose tissue (WAT) plays essential roles in whole-body metabolism, not only by storing excess energy and releasing lipids in response to energy deficits, but also by producing various adipokines and endocrine hormones. These metabolic functions of WAT require functional adaptation to the metabolic environment through the dynamic remodeling of adipose cellular composition. The remodeling capacity of WAT drastically declines with aging, with inadequate angiogenesis resulting in tissue hypoxia, chronic inflammation, and the deposition of excess extracellular matrix (ECM), ultimately leading to adipose tissue fibrosis. Adipose tissue fibrosis is a critical feature of senescence, which drastically increases with age[11,12] contributing to further adipocyte dysfunction. Adipose tissue fibrosis thereby accelerates the progression of metabolic diseases, such as fatty liver disease and type 2 diabetes mellitus[11,13]. Currently, the therapeutic options for blocking or reverting age-related adipose tissue fibrosis and dysfunction are limited. Therefore, there is a need for pharmacological treatments that can halt or reverse the aging process to mitigate adipose tissue fibrosis and its related metabolic complications.

Interestingly, recent evidence suggests that the accumulation of senescent cells in WAT is a critical process in the pathogenesis of age-associated adipose tissue fibrosis and dysfunction, as it contributes to various metabolic aberrations. Young or non-senescent (NS) adipocyte precursor cells (APCs) in WAT can give rise to mature adipocytes and brown fat-like cells (beige adipocytes) in response to high energy intake and cold exposure, respectively[14]. However, senescent APCs in aged or obese WAT lose the capacity to differentiate into mature white adipocytes (i.e., adipogenesis) or beige adipocytes; rather, the APCs exhibit increased fibro-inflammatory properties, and the adipose tissue expands through increased size of adipocytes (i.e., hypertrophy). Senescent APCs display substantially increased expression of CD9, an age-associated APC marker, and exhibit limited differentiation capacity, leading to significantly reduced hyperplastic adipogenesis[14,15]. Additionally, the age-associated functional decline in mature adipocytes and adipose immune cells plays key roles in the pathological remodeling of adipose tissue[16–18]. Notably, eliminating senescent cells by senolytics, such as dasatinib with quercetin (hereafter D + Q), or activating drug-inducible suicide genes driven by the p16 promoter, alleviates high-fat diet (HF)-induced insulin resistance in animals[19,20]. Moreover, in a recent clinical trial, a single three-day oral administration of D + Q in diabetic patients with chronic kidney disease decreased senescent cell accumulation and inflammation in the adipose tissue[21].

In this study, we tested 2150 clinically applied or FDA-approved compounds[22] to find new senotherapeutics for the treatment of age- and obesity-related metabolic diseases. We discovered that homoharringtonine (HHT), a natural plant alkaloid approved by the FDA for the treatment of chronic myeloid leukemia (CML), manifests a senotherapeutic activity in cultured APCs. Additionally, HHT treatment attenuates the age-associated phenotypes in adipose tissue of diet-induced obese mice and individuals with obesity. Furthermore, our data demonstrated that HHT treatment mitigates the aging phenotype and extends the lifespan of both progeroid and naturally aged mice. Collectively, our study suggests that HHT is a promising senotherapeutic agent for addressing age-related diseases and promoting healthy aging.

## Results

### Drug repositioning approach to discover novel senotherapeutics

We screened a library comprising 2150 clinically applied compounds to identify novel senotherapeutics among existing drugs[22]. For the primary screening, we used doxorubicin-induced prematurely senescent (PS) human dermal fibroblasts (HDFs)[23] in media containing 10% fetal bovine serum (FBS). Senescence phenotypes of PS HDFs were confirmed by SAβG staining and by assessing the protein expression

levels of p21 and p16 (Fig. 1A, S1A, and S1B). PS HDFs were incubated with each of the compounds at 100 nM for 4 days, then cell survival and cellular senescence were assessed using the cytotoxicity assay and SAβG staining, respectively. The primary screening identified 110 compounds that increased cytotoxicity, or decreased SAβG staining in PS HDFs, which were thereafter regarded as senolytic, or senomorphic candidates, respectively (Fig. 1A).

To further verify the senolytic or senomorphic effects of the 110 compounds, we conducted secondary screening by testing these compounds in multiple cell types, including replicatively senescent (RS) HDFs, RS human umbilical vein endothelial cells (RS HUVECs), and PS human retinal pigment epithelial cells (PS hRPEs) (Figs. S1C-H). Through the secondary screening of the 110 compounds, we selected 15 compounds that showed senolytic or senomorphic (hereafter senotherapeutic) activities across all three cell types (Fig. 1A, S1I, and Supplementary Table 1). ABT263 and rapamycin were used as senolytic[24] and senomorphic[25] positive controls, respectively (Fig. S1I).

### Identification of HHT as a novel senotherapeutic molecule

Adipose tissue senescence plays a causal role in the development of obesity- and age-associated insulin resistance and metabolic dysfunction[6,19]. To validate the effects of the candidate compounds on obesity and associated metabolic dysfunction, we performed pilot in vivo experiments in high-fat diet (HF)-induced obese mice. Among the 15 candidates, two were initially excluded due to known adverse effects. We excluded #1806 (trametinib) due to gastrointestinal symptoms and reduced food intake[26–28], and #231 (everolimus) due to hyperglycemia[29,30]. We tested the remaining 13 compounds in the HF-induced obese mice (Fig. 1B and Supplementary Table 2). We found that the #34 compound (HHT) mitigated weight gain and improved glucose regulation compared to PBS control in the absence of food intake differences (Fig. 1C–G and S2A–E). We then performed in vitro analyses to investigate cellular mechanisms underlying this effect in media containing 10% FBS. We found that HHT induced cytotoxicity in RS human visceral preadipocytes (HPAs) and RS HDFs in a dose-dependent manner, and elevated LDH levels in the media of these senescent cells, while reducing senescence-related phenotypes (Figs. S3A–C and S3E–G). This was accompanied by the elevation of cleaved caspase-3 (Casp-3) protein, indicating the induction of apoptotic cell death by HHT treatment in RS HPAs (Fig. S3D) and RS HDFs (Fig. S3H). Interestingly, HHT appeared to reduce cell numbers at 50 nM and 100 nM concentrations in non-senescent (NS) cells (Figs. S3A, S3B, S3E and S3F). Importantly, this did not induce LDH release (Figs. S3C and S3G), suggesting growth arrest rather than cell death, which aligns with the known mechanism of HHT in cancer cells and normal cells[31,32]. This finding was further supported by flow cytometry, which demonstrated increased G0/G1 arrest in HHT-treated NS cells (Fig. S3I). Given that HHT induces cell cycle arrest, it was necessary to clarify whether the observed decrease in NS cell numbers was due to actual cytotoxicity or simply growth inhibition. To address this, we repeated the experiments under 0% FBS conditions, where overall cell proliferation is minimized. Under these conditions, HHT induced significant cytotoxicity in senescent HPAs and HDFs while having no effect on NS cell numbers (Fig. 1H–K and S3J–M). These effects of HHT are specific to senescent cells and comparable to well-known senolytics, dasatinib+quercetin (D + Q) and ABT263 (Figs. S3N, S4, and Supplementary Table 3). Additionally, HHT treatment reduced the expression of some SASP proteins in all tested cell types, including HPAs (Fig. S5).

These results suggest that HHT mitigates glucose intolerance in mice, and exhibits senotherapeutic activity across multiple human cell types, including preadipocytes.

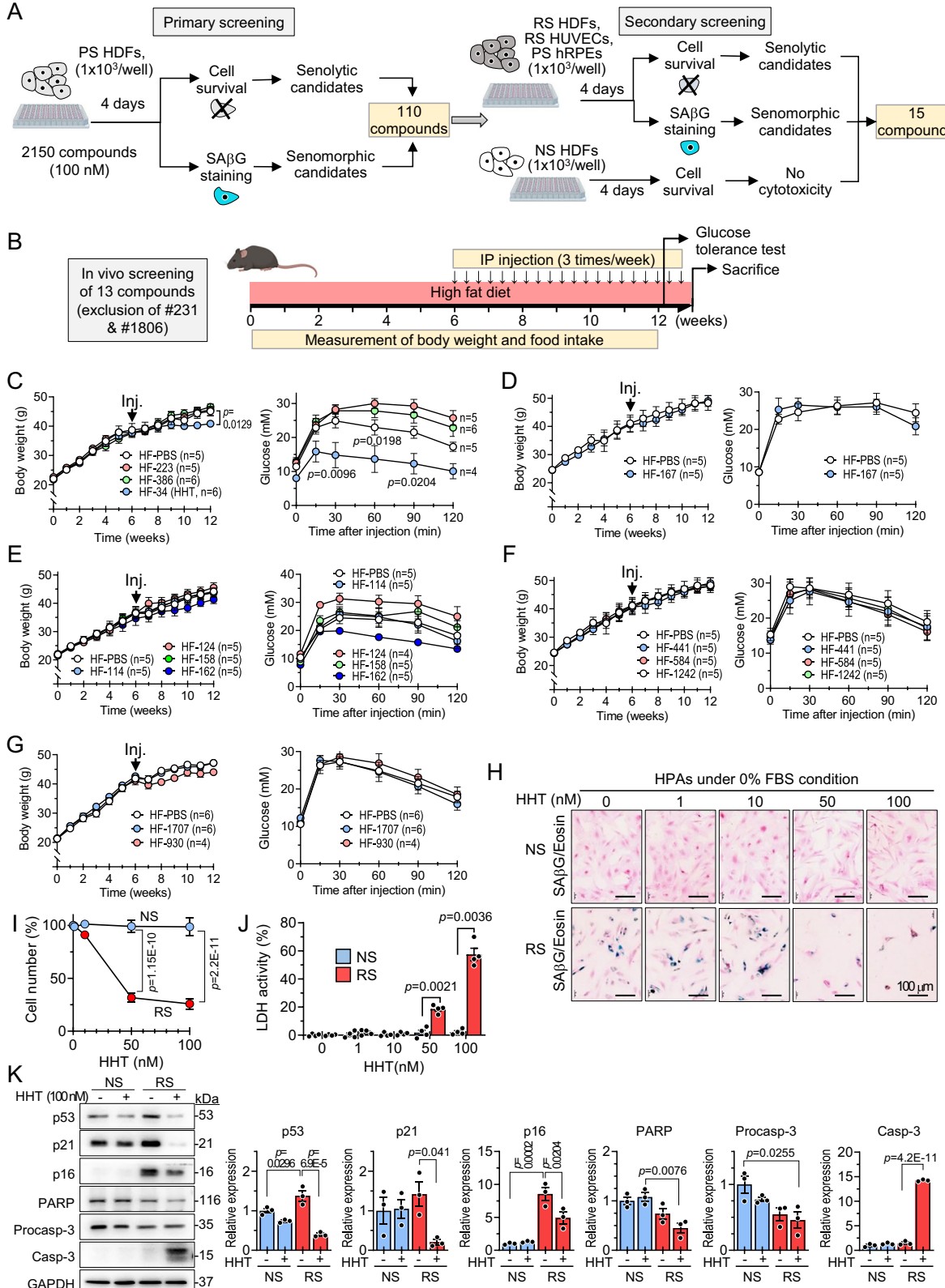

## HHT reduces adiposity and improves insulin resistance in HF-fed obese mice

To further investigate the beneficial metabolic effects of HHT on HF-induced obesity and metabolic dysfunction, we fed mice a 60% HF diet for 6 weeks, then treated with either HHT or PBS control three times weekly for 8 weeks while continuing the HF diet (Fig. 2A). Consistent with our in vivo screening studies (Fig. 1C), HHT treatment significantly reduced body weight gain from 4 weeks after the start of HHT treatment without a significant difference in accumulated food intake between the two groups (Fig. 2A, B), indicating that reduced body weight gain by HHT is not primarily attributed to decreased food intake. Importantly, we found a significant reduction in the fat weight in HHT-treated mice without significant effects on other organs (Fig. 2C and S6A), suggesting that reduced adiposity is the main

**Fig. 1 | Identification of HHT as a novel senotherapeutic. A** Screening scheme of senotherapeutic candidates using prematurely senescent human dermal fibroblasts (PS HDFs), replicatively senescent (RS) HDFs, RS HUVECs, and PS hRPEs cultured in medium with 10% FBS. Senescence in various cell types was induced by treatment with doxorubicin (Dox), defined as PS cells, and by serial cultivation, defined as RS cells, which are compared to non-senescent (NS) cells. **B** In vivo screening scheme of senotherapeutic candidates in high-fat diet (HF)-induced obese mice (n = 4 to 6). Mice were fed HF continuously for 13 weeks. After the first 6 weeks of HF feeding, they were injected with senotherapeutic candidates (3 times/week, administration dosage of each senotherapeutic in Supplementary Table 2) for the remaining 6 weeks while continuing the HF. Throughout the experimental period, body weight and food intake were measured. After completing the 12 weeks of HF feeding, an intraperitoneal glucose tolerance test (1 g/kg BW glucose) was performed. **C–G** Changes in body weight during the experimental period and glucose levels during intraperitoneal glucose tolerance test. The sample size (n) for each group is shown in the corresponding graph. NS and RS human visceral preadipocytes (HPAs) were treated with the indicated concentrations of HHT for 4 days under 0% FBS condition (**H–K**). **H** Representative images of SAβG and eosin staining. **I** Cell number measured using a CCK-8 assay (n = 4 for NS, n = 3 for RS). **J** LDH activity in media (n = 4 in each group). **K** Expression levels of age- or apoptosis-associated proteins (n = 3 in each group). Values are presented as means ± SEM and data were analyzed with one-way or two-way analysis of variance (ANOVA) followed by a post-hoc test. CCK-8 Cell Counting Kit-8; HF high-fat diet; HHT homoharringtonine; hRPEs human retinal pigment epithelial cells; HUVECs human umbilical vein endothelial cells; Inj., start injection with senotherapeutic candidate; LDH lactate dehydrogenase; SAβG senescence-associated β-galactosidase. Illustration created in part (A and B) with BioRender (https://BioRender.com/2dkuoy1).

contributor to body weight reduction. However, this effect was not observed in young mice fed a standard chow diet (Fig. S6B). Fasting plasma glucose levels were reduced by HHT (Fig. 2D) without a significant difference in fasting plasma insulin levels between the groups (Fig. 2E). Additionally, HOMA-IR was significantly lower in the HHT-treated mice than in PBS-treated mice with similar HOMA-%β level in both groups, suggesting that HHT improves HF-induced insulin resistance (Fig. 2F, G). To examine the effect of HHT on glucose homeostasis, we performed an intraperitoneal glucose tolerance test (IPGTT). The HHT-treated group showed enhanced blood glucose clearance compared to the PBS-treated group (Fig. 2H). The area under the curve (AUC) for glucose clearance was significantly lower in the HHT-treated mice compared with the PBS-treated mice (Fig. 2I). The plasma insulin levels and AUC for insulin were similar between the HHT- and PBS-treated mice (Fig. 2J, K). These results suggest that HHT reduces fat mass and improves insulin resistance in HF-fed obese mice.

## HHT reduces cellular senescence and inflammation in WAT of HF-fed obese mice

To determine whether HHT improves glucose homeostasis in HF-induced obese mice by targeting adipose tissue senescence, we evaluated senescence markers in HHT-treated white adipose tissues (WATs). HHT treatment decreased SAβG staining in various adipose depots, including subcutaneous, epididymal, mesenteric, and perirenal fat tissues, to levels similar to those in the normal chow diet control (Ch-PBS) group (Fig. 3A). Indeed, histological analysis revealed that the SAβG-positive cells in the epididymal fat tissues were significantly increased by HF, while HHT treatment significantly reduced their abundance (Fig. 3B). To further confirm the senotherapeutic activity of HHT in vivo, we employed a transgenic p16-luc mouse model that carries the entire human p16 gene locus tagged with firefly luciferase and is reported to increase bioluminescence concurrent with the p16 level with age[33]. The p16-luc mice were fed with HF for 6 weeks and then treated with HHT or PBS three times a week for 8 weeks. Notably, bioluminescence was increased in the HF (HF-PBS) compared to the Ch-PBS group, and this elevation was normalized by HHT treatment (HF-HHT, Fig. 3C). These results indicate that senescent cells accumulated in the adipose tissues of HF-fed p16-luc mice, but were cleared by HHT, demonstrating the in vivo senotherapeutic activity of HHT. Furthermore, HHT treatment significantly reduced the expression levels of p53, p21, and p16 proteins in WAT when compared with those in HF-PBS (Fig. 3D). Collectively, these data suggest that HHT attenuated cellular senescence in the adipose tissues of the HF-fed obese mice.

Senescent cells accumulate in obese WAT and stimulate the secretion of inflammatory cytokines, immune modulators, growth factors, and proteases (i.e., SASP), leading to adipose tissue dysfunction and metabolic disturbance[34]. Notably, we found that HHT treatment reduced crown-like structures (asterisks in Fig. 3E, F), suggesting the anti-inflammatory effects of HHT in WAT. HF induces hypertrophic enlargement of adipocytes, characterized by an increase in large-sized adipocytes and a decrease in small-to-medium-sized adipocytes. Treatment with HHT partially mitigated the hypertrophic expansion of the adipocytes, indicating improved WAT function (Fig. 3G). In addition, the levels of proinflammatory cytokines (IL-1β, MCP1) and their regulator (AP1)[35,36] were increased by HF, while HHT treatment suppressed this proinflammatory response (Fig. 3H). At the protein level, key components of the SASP involved in adipose tissue inflammation and fibrosis which were significantly increased by HF (TGF-β1, MMP2, and PAI-1) were reduced by HHT treatment (Fig. 3I). Protein levels of HSL and ATGL, lipolytic enzymes involved in triglyceride (TG) hydrolysis and insulin sensitivity which are known to be reduced in aged subjects[37], were also decreased by HF but normalized by HHT treatment (Fig. 3J). These results suggest that HHT attenuates HF-induced WAT inflammation and metabolic dysfunction.

To test whether HHT impacts thermogenesis, we examined the energy expenditure of the HHT-treated mice (Figs. S7A–C). We found that HHT treatment marginally (p = 0.056) increases the energy expenditure (Fig. S7C) and reduces the size of lipid droplets in brown adipose tissue (BAT) with significantly increased uncoupling protein (UCP1) expression in BAT (Figs. S7D and S7E). These data suggest that a moderate increase in energy expenditure by activation of BAT partially contributes to metabolic improvement by HHT treatment.

Taken together, these results suggest that HHT reduces adipose tissue inflammation and improves the thermogenic and lipolytic functions of adipose tissue, likely through the elimination of senescent cells in HF-fed obese mice.

## Single nucleus RNA sequencing analysis reveals that HHT attenuates adipocyte senescence in WAT of HF-fed obese mice

WAT is a highly heterogeneous endocrine organ comprising various cell populations, including mature adipocytes, APCs, immune cells, and vascular cells[38,39]. Therefore, to determine the main target cell populations that contribute to the healthy remodeling of WAT and improved metabolic homeostasis by HHT treatment, we performed single nucleus RNA sequencing (snRNA-seq) analysis[40,41]. The epididymal WAT of three mice from each group were pooled and processed. Unsupervised clustering based on differentially expressed genes (DGEs) revealed distinct WAT cell populations that are projected onto 14 different clusters in both PBS- and HHT-treated adipose tissues, as illustrated in the t-distributed stochastic neighbor embedding (tSNE) plots (Fig. 4A). Similar to previous reports[40,41], these include mature adipocytes, APCs, macrophages, T- and B-cells, and vascular endothelial cells. The heat map for the top three genes of each cluster demonstrates the cell-type-specific transcriptome signature (Fig. S8A). The stacked bar plot for the proportion of each cluster demonstrates the impact of HHT treatment on the frequency of each cell population (Fig. 4B). We confirmed the expression of cell-type-specific genes for major cell populations, including adipocytes (Cluster 5), APCs (Cluster

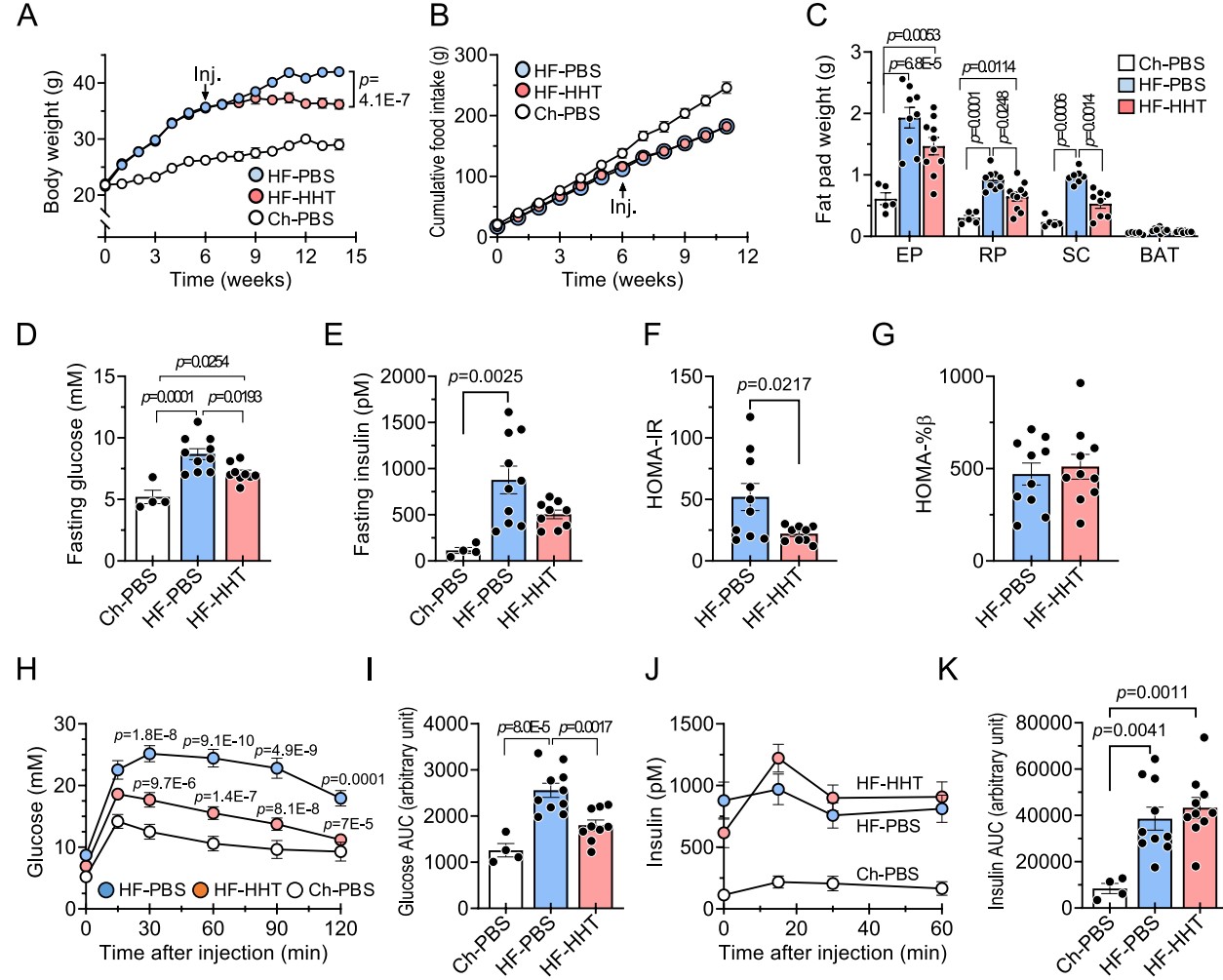

**Fig. 2 | HHT prevents weight gain and improves insulin resistance in HF-fed mice.** After feeding a high-fat diet (HF) for 6 weeks, male mice were administered with HHT or PBS for 8 weeks while they were kept on HF. Some animals were kept on a normal chow diet and treated with PBS. **A** Body weight changes ($n = 5$ for Ch-PBS; $n = 10$ for HF-PBS and HF-HHT). **B** Accumulated food intake ($n = 4$ for Ch-PBS; $n = 10$ for HF-PBS and HF-HHT). **C** Fat pad weight ($n = 5$ for EP, RP, SC, and BAT in the Ch-PBS group; $n = 10$ for EP, RP, and BAT in the HF-PBS and HF-HHT groups; $n = 8$ for SC in the HF-PBS and HF-HHT groups). **D** Fasting glucose ($n = 4$ for Ch-PBS; $n = 10$ for HF-PBS; $n = 9$ for HF-HHT). **E** Fasting insulin ($n = 4$ for Ch-PBS; $n = 10$ for HF-PBS; $n = 9$ for HF-HHT). **F** Homeostatic model assessment for insulin resistance (HOMA-IR) ($n = 9$ for HF-HHT; $n = 10$ for HF-PBS). **G** HOMA for β-cell function (HOMA-%β) ($n = 10$ per group). **H, I**. Blood glucose levels and their area under the curve (AUC) during intraperitoneal glucose tolerance test ($n = 4$ for Ch-PBS; $n = 10$ for HF-PBS; $n = 9$ for HF-HHT). $p$ values were determined for the comparisons between HF-PBS and HF-HHT, and between Ch-PBS and HF-PBS. **J, K** Plasma insulin levels and their AUC during intraperitoneal glucose tolerance test ($n = 4$ for Ch-PBS; $n = 10$ for HF-PBS and HF-HHT). Values are presented as means ± SEM. Data were analyzed via one-way or two-way ANOVA followed by a post-hoc test when comparing more than two groups or two-tailed Student's $t$ test when comparing two groups. BAT, brown adipose tissue; Ch chow diet; EP epididymal fat; HF, HHT homoharringtonine; PBS phosphate-buffered saline; RP retroperitoneal fat; SC subcutaneous fat.

2), mesothelial cells (Cluster 8), macrophages (Cluster 0, 1, 3, 7), T-cells (Cluster 4), and B-cells (Cluster 6) (Fig. S8B).

Notably, the proportion of adipocytes (Cluster 5) increased in the HF-HHT-treated WAT compared with the HF-PBS-treated WAT (Fig. 4A, B). This result, along with reduced adipocyte sizes (Fig. 3E, G), suggests increased hyperplastic adipogenesis (i.e., adipocyte cell number increase), which is closely associated with improved WAT metabolic function and enhanced glucose homeostasis. To gain insights into the HHT-mediated molecular changes in adipocytes, we performed DGE analysis between the HF-HHT- and HF-PBS-treated adipocytes and found significantly altered genes presented in the volcano plot (Fig. 4C middle panel). Subsequent gene ontology (GO) enrichment analysis of DGEs in the HHT-treated adipocytes exhibited enriched signatures associated with insulin sensitivity (e.g., *Adipor2*), lipid and carbohydrate catabolic process, and fat cell differentiation (Fig. 4C, right panel; HHT > PBS plot). In contrast, the PBS-treated adipocytes showed

signatures of cell migration and tissue organization (Fig. 4C, left panel; PBS > HHT plot). We further characterized the functional aspect of mature adipocytes (Cluster 5) and found a positive correlation of HHT-treated adipocytes with healthy adipokine profiles, such as *Adipoq* (Adiponectin), *Cfd* (Adipsin), and *Vegfa* (VEGF-A) (Fig. 4D)[42–45]. Next, to directly test whether HHT treatment reduces adipose tissue senescence, we analyzed the SenMayo gene set, a panel of 125 genes that are associated with cellular senescence[46]. We examined the SenMayo signature scores in various cell types and observed that the score was significantly lower in adipocytes and myeloid/erythroid cells, while it was increased in macrophages with higher M1-like and lower M2-like signature (Fig. 4E, S8C and S8D). This finding is unexpected and somewhat inconsistent with our histopathological data, which showed a reduced crown-like structure (CLS) following HHT treatment (Fig. 3E). Despite the seemingly contradictory transcriptomic signatures, our immunofluorescent staining with the M1-like macrophage

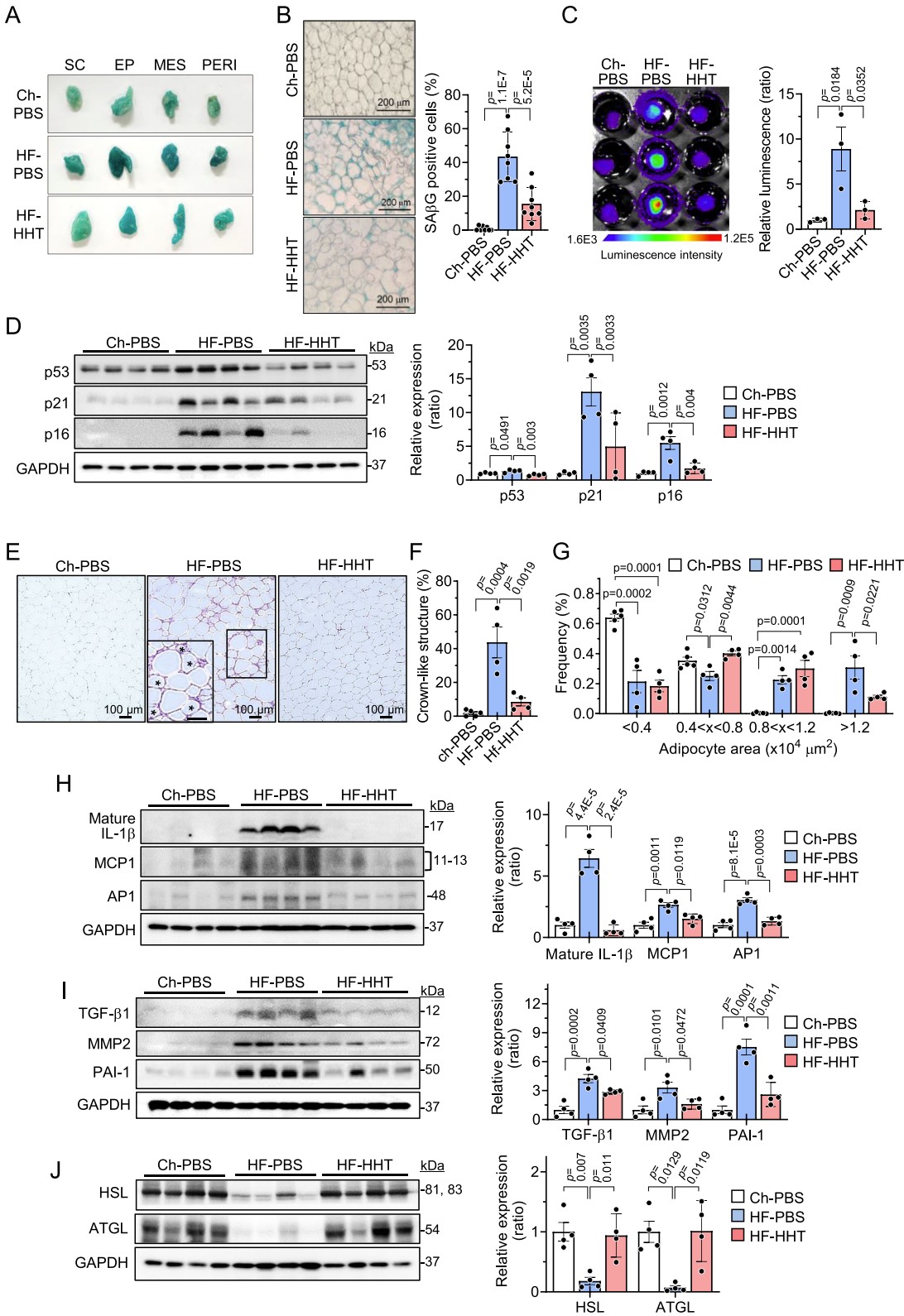

marker iNOS further confirmed that CLS with iNOS-positive pro-inflammatory macrophages was significantly reduced in HHT-treated WAT (Fig. S8E). These findings indicate that HHT attenuates adipocyte senescence and may exert differential effects across various cell types. Furthermore, HHT-treated adipocytes exhibited significantly lower γ-H2AX immunofluorescent signal, indicating lower levels of DNA damage (Fig. 4F). Taken together, these data illustrate that HHT suppresses adipocyte senescence and improves the metabolic function of adipocytes.

## HHT improves APC plasticity and metabolic function in WAT of HF-fed obese mice

Our snRNA-seq demonstrated an increase in adipocyte number in HHT-treated adipose tissue (Fig. 4A, B). Increased adipocyte number is

**Fig. 3 | HHT attenuates adipose tissue senescence in HF-fed obese mice.** After feeding high-fat diet (HF) for 6 weeks, male mice were administered with HHT (HF-HHT) or PBS (HF-PBS) for 8 weeks while they were kept on HF. Chow diet-fed mice were kept on a chow diet during the experimental period and injected with PBS for 8 weeks (Ch-PBS). **A** Senescence-associated β-galactosidase (SAβG) staining in various adipose tissues. **B** Histological analysis of SAβG stained epididymal white adipose tissue (WAT) ($n = 8$ in each group). Scale bar, 200 μm. **C** p16-luciferase mice were fed with HF for 6 weeks and treated with HHT for 8 weeks while kept on HF. Bioluminescence intensity of the epididymal WAT ($n = 3$ in each group). **D** The levels of p53, p21, and p16 proteins in the epididymal WAT ($n = 4$ in each group). **E** Histological analysis of epididymal WAT after staining with hematoxylin and eosin. The adipocytes marked with asterisks were the cells surrounded by macrophages which counted as positive crown-like structures. Scale bar, 100 μm. **F** The percentages of crown-like structure positive cells in the epididymal WAT ($n = 5$ in

chow diet group and $n = 4$ in each HF group). **G** The size distribution of adipocytes in the adipose tissues ($n = 5$ in chow diet group and $n = 4$ in each HF group). **H** Expression levels of IL-1β, MCP1, and AP1 proteins in the epididymal WAT ($n = 4$ in each group). **I** Expression levels of TGF-β1, MMP2, and PAI-1 proteins in the epididymal WAT ($n = 4$ in each group). **J** Expression levels of HSL and ATGL proteins in the epididymal WAT ($n = 4$ in each group). Values are presented as means ± SEM. Data were analyzed via one-way ANOVA followed by a post-hoc test. AP1 activator protein 1; ATGL adipose triglyceride lipase; Ch chow; EP epididymal fat; GAPDH glyceraldehyde 3-phosphate dehydrogenase; IL-1β interleukin-1β; HHT homoharringtonine; HSL hormone sensitive lipase; MCP1 monocyte chemoattractant protein-1, MES mesenteric fat; MMP2 matrix metalloproteinases 2, PAI-1 plasminogen activator inhibitor-1; PBS phosphate buffered saline; PERI perirenal fat; SC subcutaneous fat; TGF-β1 transforming growth factor-β1.

directly linked with the improved metabolic function of WAT and is mediated through the expansion and differentiation of APCs, which are significantly reduced in age-associated WAT[47,48]. To explore the effect of HHT on APCs, we performed DGE analysis between the HF-HHT- and HF-PBS-treated APCs and found an increased signature of ECM organization and tissue adhesion in the APC population in the PBS-treated APCs (Cluster 2) (Fig. 5A, left panel: PBS > HHT plot). The HF-PBS-treated APC showed enriched expression of collagen genes (*Col5a2* and *Col4a1*) compared with the HF-HHT-treated APCs (Fig. 5A, middle volcano plot). Our GO analysis revealed that the collagen biosynthesis process was significantly reduced by HHT treatment in several cell clusters, including adipocytes (Cluster 5) and APCs (Cluster 2), two major players of adipose fibrosis (Fig. 5B)[15,49]. Notably, we found that HHT significantly reduces the intensity of fibrosis-specific staining (Fig. 5C). Since adipose fibrosis is closely related to a decline of the adipogenic differentiation of APCs[14,15], we tested the effect of HHT on the differentiation of APCs using a senescent 3T3-L1 differentiation model (Fig. S9A). Senescent 3T3-L1 preadipocytes lost their adipogenic differentiation capacity, and HHT treatment restored this capability, as confirmed by oil red O (ORO) staining (Fig. S9B). Moreover, HHT treatment reduced senescence phenotypes, such as reduced SAβG-positive cells and p16 expression, of 3T3-L1 cells (Figs. S9C and S9D). Together, our results demonstrate that HHT treatment enhances the adipogenic differentiation of APCs. APCs' cellular plasticity, such as cold-induced beige adipocyte formation, has been shown to decline with age[14,50]. To investigate the effect of HHT on cold-induced beige adipogenesis, we kept the mice in a cold (4 °C) environment (Fig. 5D). The expression of cold-induced beige-specific genes was significantly increased in HHT-treated mice compared with the PBS-treated group (Fig. 5E). This indicates that HHT restores the potential of APCs to form cold-induced beige adipocytes, which is otherwise lost in metabolically unhealthy adipose tissue[14].

Finally, to test whether the improved whole-body glucose handling observed with HHT treatment is a result of improved metabolic function of the adipose tissue, we performed hyperinsulinemic-euglycemic clamp studies. HHT administration restored the whole-body glucose turnover that was decreased by HF (Fig. 5F). We also found increased glucose uptake in the HHT-treated WAT compared with the PBS-treated WAT (Fig. 5G), indicating enhanced insulin sensitivity in WAT following HHT treatment. Additionally, we examined liver tissues and found significantly reduced steatosis in the HHT-treated liver with decreased TG accumulation and alanine aminotransferase (ALT) enzyme (Fig. 5H–J). These findings indicate that HHT-mediated enhancement of APC plasticity, along with reduced adipocyte senescence, promotes healthier adipose tissue remodeling and metabolic homeostasis in HF-fed obese mice.

## HHT improves adipose tissue senescence and insulin resistance in aged mice

Next, to examine the effect of HHT on age-associated insulin resistance and adipose tissue dysfunction, aged mice (18 months) fed with a

normal chow diet were administered PBS or HHT three times a week. Body weight and accumulated food intake were not significantly different between the groups (Fig. 6A, B). However, the cumulative weight gain was significantly lower in the HHT-treated mice, leading to a reduced food efficiency ratio in the HHT group (Fig. 6C, D). Importantly, HHT treatment improved blood glucose clearance and insulin sensitivity (Fig. 6E–H) in the aged mice. Furthermore, the percentages of SAβG-positive cells and adipocyte size were reduced in the HHT-treated WAT (Fig. 6I and S10A). Additionally, the levels of p21 and p16 proteins were significantly reduced in HHT-treated WAT compared to PBS-treated WAT (Fig. 6J). The levels of precursor IL-1β, p-p65/p65 ratio, TGF-β1, MMP2, and PAI-1 proteins were also significantly reduced by HHT treatment (Fig. 6K and S10B). The biological age of WAT in HHT-treated mice was also assessed using epigenetic age and was observed to be lower in HHT-treated mice (Fig. 6L). These results suggest that HHT attenuates adipose tissue senescence and improves metabolic disturbance in aged mice.

We showed the senotherapeutic effect of HHT in human pre-adipocytes, suggesting its potential impact on human WAT. Thus, we further tested the effect of HHT on cellular senescence ex vivo in human WAT. As expected, HHT treatment decreased SAβG staining in human WAT in a dose-dependent manner (Fig. 6M). Consistent with this, HHT decreased the SAβG-positive cells in the tissue sections of the human WAT samples (Fig. S10C) and the levels of p53 and p21 proteins (Fig. 6N) in a dose-dependent manner. These findings indicate that HHT has senotherapeutic activity in human adipose tissue.

## HHT exerts senotherapeutic effects by targeting HSPA5

To explore a potential target of HHT, we employed a Drug Affinity Responsive Target Stability (DARTS) approach[51] followed by mass spectrometry. DARTS is based on the principle that the binding of a small molecule to a target protein stabilizes the protein's structure and makes it protease resistant (Fig. 7A). We initially performed DARTS using RS HDFs and observed many proteins that are resistant to proteinase (i.e., pronase) upon HHT treatment (Fig. S11A). Among them, we selected 25 pronase-resistant protein bands on several gels, which might bind putatively with HHT. With mass spectrometry, we then identified 12 candidate proteins (Supplementary Table 4). We further confirmed by Western blot whether these 12 candidate proteins are resistant to pronase through binding with HHT, and identified the stability of 5 proteins: PKM2, HSPA5, HSPA8, TXNRD1, and RPL7, upon HHT treatment (Fig. 7B and S11B). To identify the target proteins that mediate the senotherapeutic activity of HHT, each candidate protein was knocked down in RS HPAs with siRNAs (Fig. S11C), and then the senotherapeutic activity of HHT was tested. Among them, the knockdown of HSPA5 (heat shock protein family A member 5) abolished HHT-induced cytotoxicity, which was accompanied by reduced Casp-3 levels (Fig. 7C–E). Conversely, we overexpressed HSPA5 in HPAs and found that the senotherapeutic activity of HHT was enhanced in HSPA5 overexpressed cells (Fig. 7F and S11D). HSPA5 protein levels were

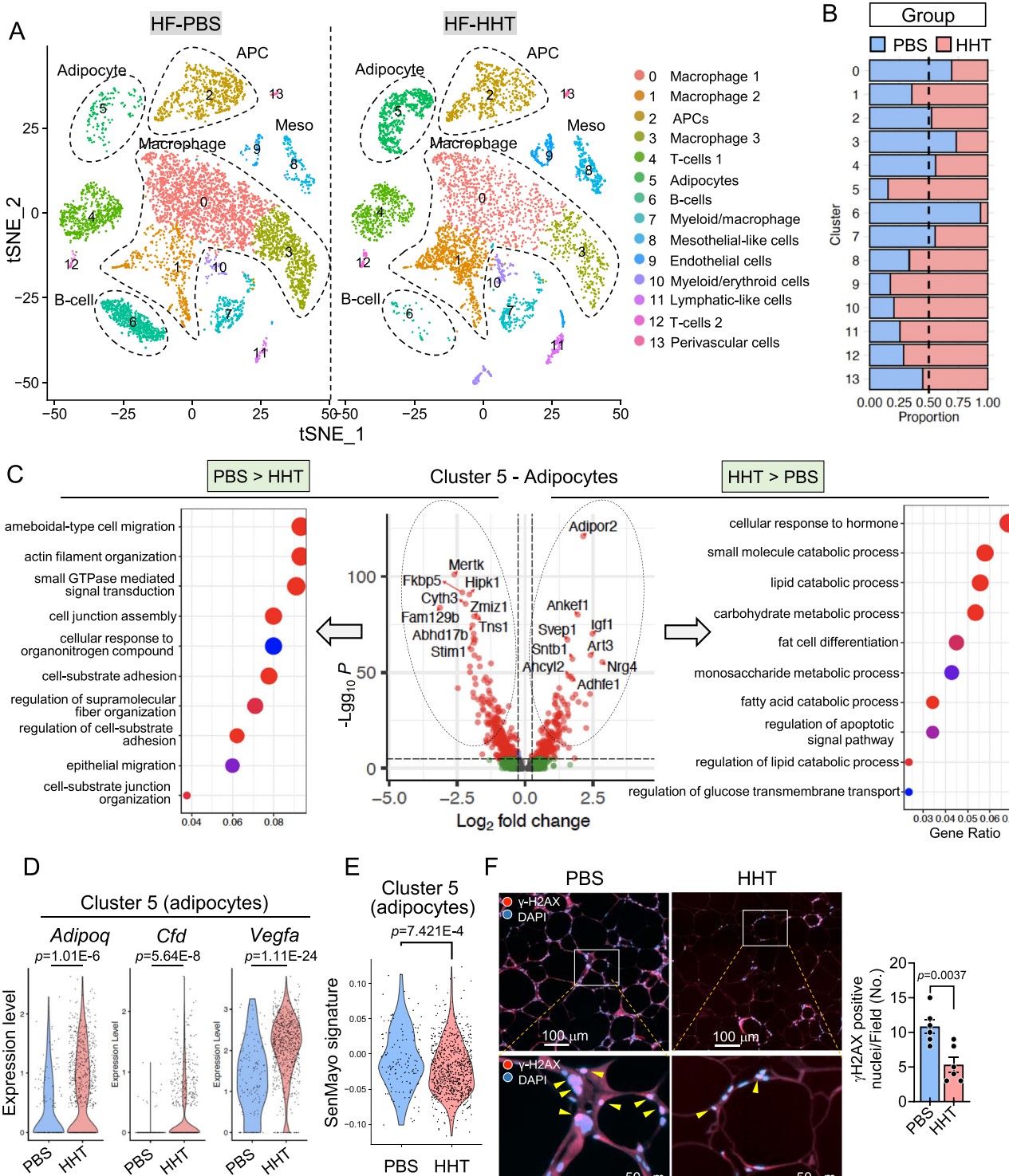

**Fig. 4 | HHT treatment induces global alterations in adipose tissue cell populations and reduces the senescence of adipocytes.** After feeding high-fat diet (HF) for 6 weeks, male mice were administered with HHT or PBS for 8 weeks while they were kept on HF. Pooled epididymal fat was analyzed with single nucleus RNA sequencing (snRNA-seq). **A** t-distributed stochastic neighbor embedding (tSNE) plot of snRNA-seq analysis showed multiple adipose tissue cell types. **B** Proportion of each cell population in PBS or HHT-treated adipose tissue. **C** Gene ontology enrichment analysis of differentially expressed genes (DEGs) in adipocytes (Cluster 5). Differential gene expression between conditions was tested using Seurat's MAST hurdle model using a likelihood ratio test, and $p$ values (middle panel) were Bonferroni-adjusted across all genes. Differential gene ontology enrichment was run using a one-sided test, and $p$ values were corrected using FDR correction. **D** Healthy adipokine profile of mature adipocytes (Cluster 5). **E** SenMayo signature module score in adipocytes (Cluster 5) from HHT- or PBS-treated adipose tissue (Wilcoxson test). **F** Immunofluorescence staining of γ-H2AX and quantification of γ-H2AX positive nuclei per field in epididymal fat tissues ($n = 6$ in each group). Values are presented as means ± SEM. Data were analyzed via two-tailed Student's $t$ test. DAPI, 4',6-diamidino-2-phenylindole; γ-H2AX, H2A histone family member X; HF, high-fat diet; HHT, homoharringtonine; PBS, phosphate-buffered saline.

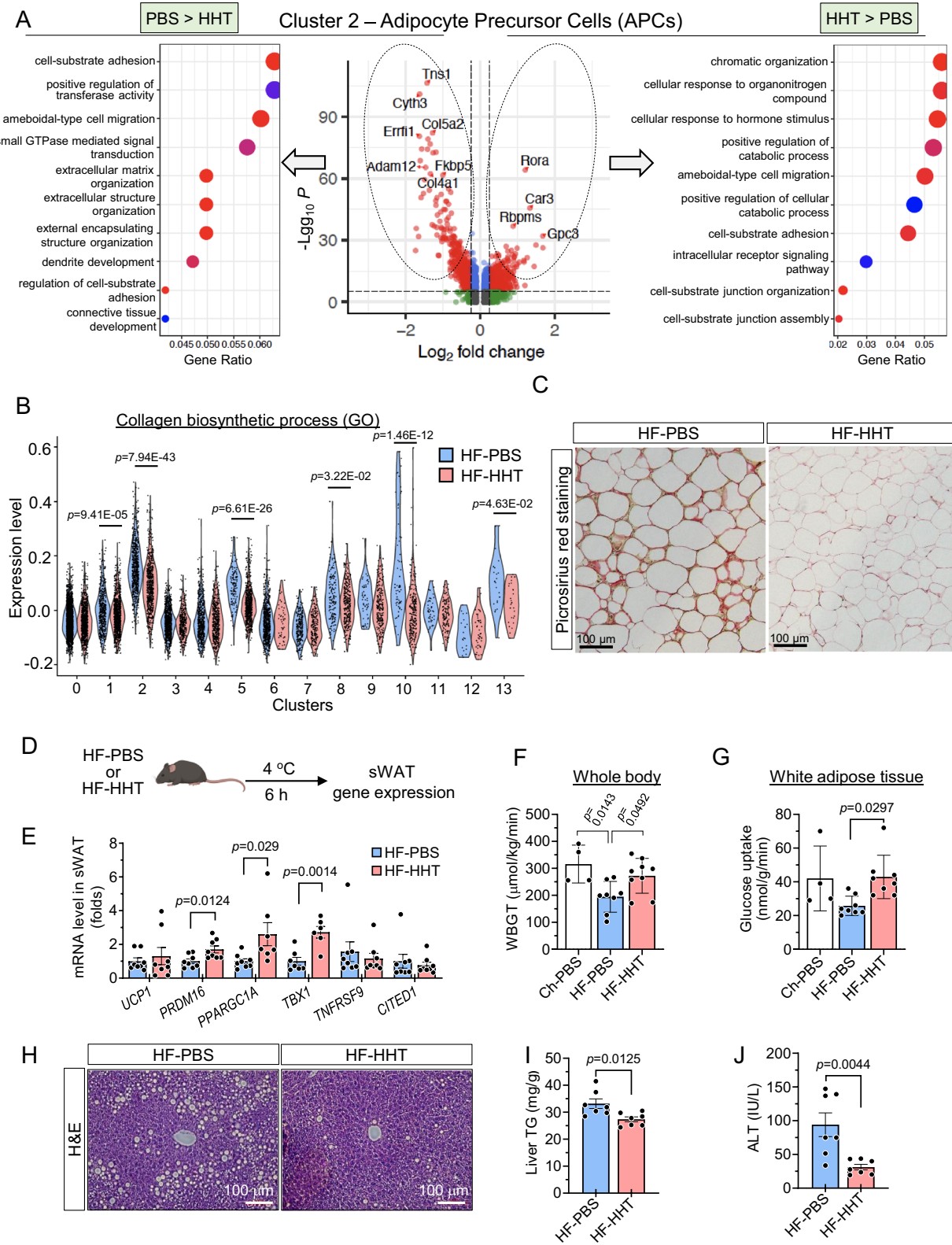

consistently higher in senescent cells than in NS cells across HPAs, HDFs, and hRPEs (Fig. S11E). To further confirm whether the senotherapeutic activity of HHT might be mediated through HSPA5, NS and RS HPAs were treated with 100 nM HHT for 2 days–a time window selected to capture the induction and progression of apoptosis without causing overt cell death–and subsequently subjected to co-immunostaining for HSPA5 and Casp-3, using DAB (brown) and Fast

Red (red) as chromogens, respectively. RS HPAs treated with HHT consistently showed higher expression of HSPA5 and Casp-3 than HHT-treated NS HPAs (Fig. 7G). Among HHT-treated RS HPAs, cells with high HSPA5 expression showed a significantly greater number of Casp-3-positive cells compared with those expressing low HSPA5 (Fig. 7H). These data suggest that HSPA5 high expressing cells are primary target of HHT, and HSPA5 may serve as a key mediator of the senotherapeutic

**Fig. 5 | HHT treatment reduces fibro-inflammatory signatures in adipocyte precursor cells and improves insulin resistance.** Male mice were fed a high-fat diet (HF) for 6 weeks and then treated with HHT or PBS for 8 weeks while remaining on HF. Pooled epididymal white adipose tissue (eWAT) was analyzed by single nucleus RNA sequencing (snRNA-seq). **A** Gene ontology enrichment analysis of differentially expressed genes (DEGs) adipocyte precursor cells (APCs, Cluster 2). DEGs were identified using Seurat's MAST hurdle model with a likelihood ratio test, and *p* values (middle panel) were Bonferroni-adjusted across all genes. Gene ontology enrichment was performed using a one-sided test with false discovery rate (FDR) correction. **B** Module score analysis of genes associated with collagen biosynthetic process across all clusters. **C** Fibrosis-specific (Picrosirius red) staining of eWAT frm PBS- or HHT-treated mice. **D** Male mice were exposed to 4 °C for 6 h with ad libitum access to food and water, after which subcutaneous WATs (sWATs) were collected for gene expression analysis. **E** Expression of browning-related genes in ($n = 8$ for Ucp1, Prdm16, Tnfrsf9, and Cited1 in both HF-PBS and HF-HHT groups, and Ppargc1a in HF-PBS group; $n = 7$ for Tbx1 in HF-PBS and Ppargc1a in HF-HHT; $n = 6$ for Tbx1 in HF-HHT). Whole-body and eWAT insulin sensitivity were assessed by hyperinsulinemic-euglycemic clamp in HF-fed obese mice treated with HHT (**F**–**G**). **F** Whole body glucose turnover (WBGT) ($n = 4$ for Ch-PBS, $n = 8$ for HF-PBS, $n = 9$ for HF-HHT). **G** Glucose uptake in eWAT ($n = 4$ for Ch-PBS, $n = 8$ for each HF group). **H** Liver histology assessed by hematoxylin and eosin staining. **I** Hepatic triglyceride levels ($n = 7$ per group). **J** Plasma alanine aminotransferase (ALT) levels ($n = 7$ per group). Data are presented as means ± SEM and analyzed by one-way ANOVA followed by a post-hoc test when comparing more than two groups or two-tailed Student's *t* test when comparing two groups. Ch, chow diet; CITED1, Cbp/P300 interacting transactivator with Glu/Asp rich carboxy-terminal domain 1; HHT, homoharringtonine; PBS, phosphate-buffered saline. Illustration created in part (D) with BioRender (https://BioRender.com/y2s7wj2).

activity of HHT. We also prepared HPAs in three distinct states of cellular senescence through serial passaging: low passage (p2-p3), mid passage (p6-p7), and high passage (p12-p13). As passage number increased, the expression of HSPA5, p53, p21, and p16 proteins increased and the senotherapeutic effect of HHT was enhanced with the elevated expression of HSPA5 (Figs. 7I, J, and S11F–H).

To directly assess the physical interaction between HHT and HSPA5, we conducted a Surface Plasmon Resonance (SPR) assay, which allows for the measurement of HHT's affinity for HSPA5 protein[52]. The SPR assay revealed a significant molecular affinity between HHT and HSPA5 (Fig. 7K). To further investigate the molecular binding between HHT and HSPA5, we employed a binding model prediction. The initial binding model was determined using rigid docking, followed by Molecular Mechanics energies combined with the Generalized Born and Surface Area (MM-GBSA) calculation, a widely used method for estimating the free energy of the binding of small molecules to protein[53]. HSPA5 is known to play a pivotal role in cell survival, and suppression of HSPA5 by inhibitors that bind to the ATP binding site of HSPA5 can lead to cell death[54]. Therefore, we postulated that HHT may inhibit the function of HSPA5 through binding to the ATP binding site. The predicted binding model of HSPA5 and HHT shows key interactions including four hydrogen bonds (H-bonds) and one pi-cation interaction with HSPA5 residues (Fig. 7L and S11I). These results suggest that HHT can bind to HSPA5 and may act as an inhibitor of HSPA5. To test this hypothesis, we measured the ATPase activity of HSPA5 in the presence of HHT or HA15, a potent and specific inhibitor of HSPA5[55]. We found that the inhibitory activity of HHT was greater than HA15, demonstrating that HHT acts as an HSPA5 inhibitor by reducing its ATPase activity (Fig. 7M). Our data collectively suggest that HHT physically interacts with HSPA5 to inhibit its ATPase activity and induces cytotoxicity in HSPA5 high-expressing senescent cells, subsequently leaving HSPA5 low-expressing cells.

In metabolic tissues, HSPA5 protein levels were significantly increased in both adipose tissue and skeletal muscle from HF-induced obese mice compared with chow-fed controls, whereas its expression was not significantly changed in the liver (Fig. 7N and S11J). The increase in HSPA5 expression was more prominent in adipose tissue than in skeletal muscle, suggesting that adipose tissue is a primary metabolic target organ responsive to HHT treatment. Indeed, HHT treatment in HF-fed obese or aged mice resulted in reduced levels of HSPA5 protein in adipose tissues due to the elimination of HSPA5 high-expressing senescent cells (Fig. 7N–P), providing mechanistic insight into the senescent cell targeted activity of HHT. In addition to its action through HSPA5, HHT is also known to inhibit the translation of ribosomal protein[56], which may be involved in mediating the senotherapeutic effect of HHT. Thus, to test this possibility, we examined the expression of ribosomal protein, RPS14, which is known to induce cell cycle arrest and senescence[57]. No significant alteration in the level of

RPS14 protein was observed in either non-senescent or senescent cells treated with HHT (Fig. S12A). In addition, we tested whether other translation inhibitors, such as cycloheximide (CHX), anisomycin (Aniso), and puromycin (Puro), influence the senescence of RS HPAs. We observed that none of the three translation inhibitors altered cell numbers, expression of aging biomarkers, and SAβG staining in RS HPAs (Figs. S12B–D). These findings indicate that the senotherapeutic activity of HHT is not associated with its translation-inhibitory function in HPAs. Taken together, these data suggest that HSPA5 may be the target molecule mediating the senotherapeutic activity of HHT, independent of HHT's ribosomal protein inhibition activity.

## HHT improves aging phenotypes and extends lifespan in progeroid mice

To further evaluate the effect of HHT on aging phenotypes and lifespan, we employed a *Zmpste24*[-/-] progeroid mice, an animal model for Hutchinson-Gilford progeria syndrome[58]. HHT administration improved physical activity in *Zmpste24*[-/-] mice (Supplementary Movie 1). HHT treatment significantly extended lifespan, increasing the median lifespan from 14 to 15 weeks and the maximum lifespan from 15 to 19 weeks (Fig. 8A). HHT also increased bone mineral content, bone area, lean body mass (Figs. 8B, C, and S13A), and grip strength (Fig. 8D) in *Zmpste24*[-/-] mice. To determine whether HHT administration improves pulmonary function, a key aspect of the senescent phenotype in this model[59], we conducted non-invasive whole-body plethysmography and found that airway resistance measured by enhanced pause (Penh) was significantly decreased in *Zmpste24*[-/-] mice (Fig. 8E). Histological analysis revealed that HHT treatment decreases collagen deposition (Fig. 8F), the percentages of SAβG staining, and p53 positive cells (Fig. 8G) in the lung tissues of *Zmpste24*[-/-] mice. In addition, HHT reduced the levels of senescence-related proteins (p53, p21, and p16), which are increased in the lung tissues of *Zmpste24*[-/-] mice (Fig. 8H). These results suggest that HHT treatment mitigates aging phenotypes in progeroid mice.

Mice often develop lymphoma, which is among the leading causes of death in aged mice[60,61]. As HHT has been used for the treatment of hematological oncology, the extended lifespan may primarily be due to the anti-cancer effects of HHT rather than its senotherapeutic effects. However, we did not find any differences in blood smear (Fig. S13B), CBC profiles (Supplementary Table 5), the numbers of Peyer's patches (Fig. S13C), and the weight of inguinal lymph nodes (Fig. S13D) among *Zmpste24*[+/+], *Zmpste24*[+/-], and *Zmpste24*[-/-] mice treated with HHT or vehicle (PBS) control. Additionally, no significant histological alterations were observed in the Peyer's patches, inguinal lymph nodes, or the spleen (Fig. S13E). These results suggest that HHT mitigates aging phenotypes and extends lifespan in the absence of tissue toxicity in progeroid mice, independent of lymphocyte suppression.

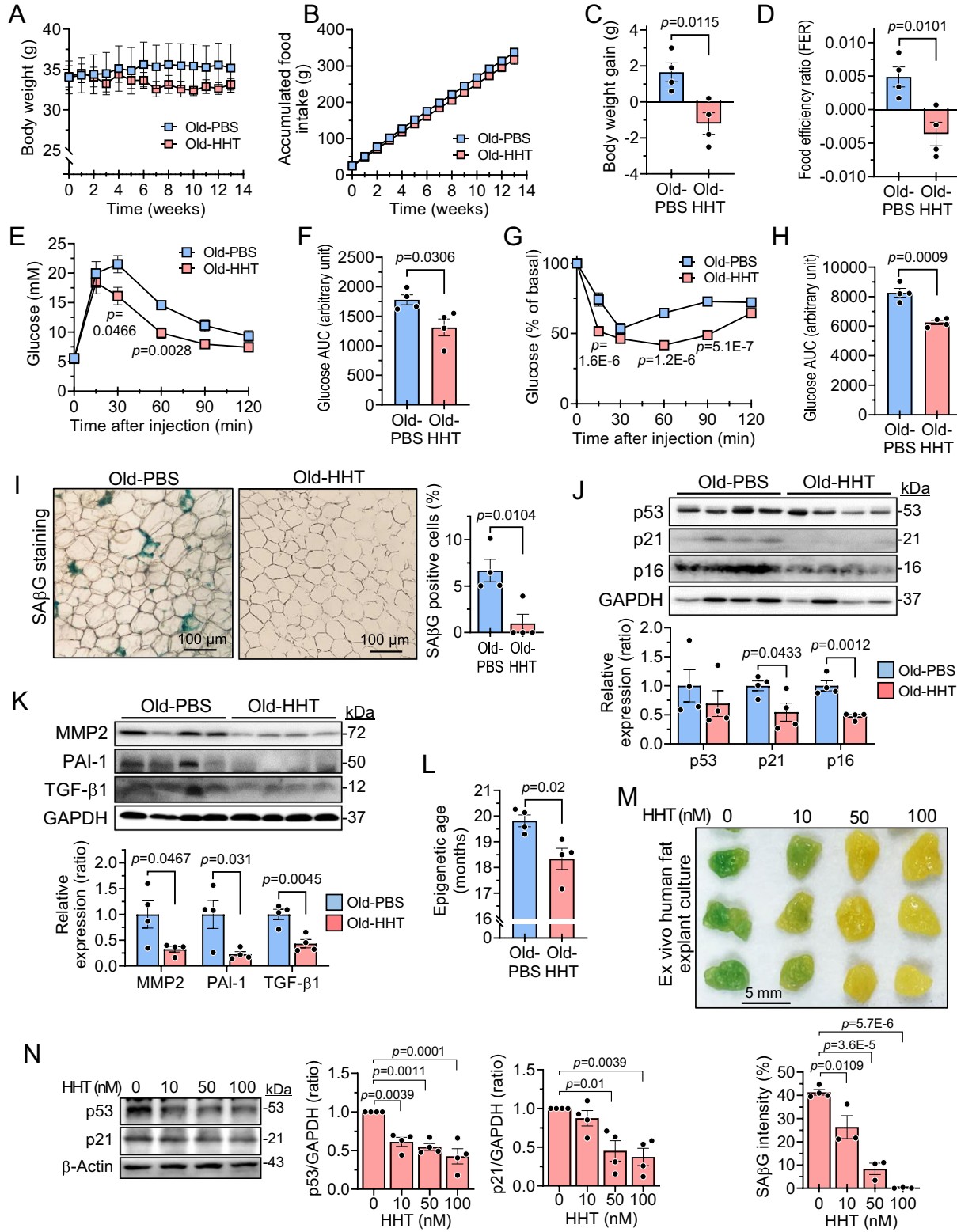

## HHT delays aging and extends lifespan in aged mice

Given these beneficial effects of HHT on metabolic function in the aged mice, we postulate that HHT also impacts general health and longevity of aged mice. To test this, we treated 16-month-old mice with HHT once per week over a period of 12 months (Fig. 9A). We found that HHT significantly extended the lifespan and increased physical activity of aged mice (Fig. 9B and Supplementary Movie 2). This was accompanied by improved kidney function, represented by reduced

proteinuria (Fig. 9C, D). It also improved skeletal muscle function measured by grip strength (Fig. 9E). In addition, HHT reduced fibrosis of the kidney, lung and liver in aged mice (Fig. S14A). HHT is primarily metabolized by the liver and excreted in the urine through the kidneys[62,63]. Thus, to test the potential adverse effects of prolonged treatment with HHT on these organs, we measured the levels of liver enzymes (i.e., aspartate aminotransferase (AST) and ALT), creatinine, and urea nitrogen (UN). We did not find any differences between PBS-

**Fig. 6 | HHT improves insulin resistance and adipose tissue senescence in aged mice and reduces senescence of human adipose tissue.** Eighteen-month-old mice were administered with HHT or PBS three times a week for 13 weeks (A-L) ($n = 4$ in each group). Body weight and food intake were measured manually once a week during HHT administration. **A** Body weight. **B** Accumulated food intake. **C** Body weight gain. **D** Food efficiency ratio. **E** Blood glucose levels during intra-peritoneal glucose tolerance test (IPGTT). **F** Area under the curve (AUC) for glucose in IPGTT. **G** The percent of basal glucose during intraperitoneal insulin tolerance test (IPITT). **H** AUC for glucose in IPITT. **I** Histological analysis of epididymal white adipose tissue (WAT) after senescence-associated β-galactosidase (SAβG) staining, and quantification of SAβG positive cell numbers. Scale bar, 100 μm. **J**. Expression levels of p53, p21, and p16 proteins in epididymal WAT. **K** Expression levels of

MMP2, PAI-1, and TGF-β1 proteins in the epididymal WAT. **L** Epigenetic age assessed using DNA methylation in retroperitoneal WAT. Human subcutaneous WAT was ex vivo cultured and treated with the indicated concentrations of HHT for 4 days (M-N) ($n = 4$ in each group). **M** SAβG staining and quantification of SAβG density ($n = 4$ for 0 nM HHT; $n = 3$ for all other groups). Scale bar, 5 mm. **N** Expression levels of p53 and p21 proteins ($n = 4$ in each group). Values are presented as means ± SEM. Data were analyzed via one-way or two-way ANOVA followed by a post-hoc test or via two-tailed Student's $t$ test. GAPDH glyceraldehyde 3-phosphate dehydrogenase; HHT homoharringtonine; MMP2 matrix metalloproteinases 2, PAI-1 plasminogen activator inhibitor-1; PBS phosphate-buffered saline; TGF-β1 transforming growth factor-β1.

and HHT-treated groups (Figs. S14B-G). Furthermore, as suppression of hematopoiesis is observed in HHT-treated cancer patients[63,64], we examined the number of circulating hematopoietic cells and spleen histology, and again found no difference between control and long-term HHT-treated mice (Fig. S14H and Supplementary Table 6). Collectively, our data demonstrate that HHT delays aging and extends lifespan in mice without major adverse effects.

As obesity-induced metabolic impairment expedites the aging process and exacerbates age-related diseases[65,66], we employed an additional aging model by a combination of HF and aging. Two-month-old mice were fed a HF diet for 11 months, and HHT was administered by once-weekly injections for 5 months while continuing the HF regimen (Fig. 9F). We found that HHT significantly extended the lifespan of mice, increasing the median lifespan from 509 to 551 days, whereas the maximum lifespan could not be determined because the mice were sacrificed at 558 days for tissue collection (Fig. 9G). In addition, HHT induced weight loss, mainly attributable to fat rather than lean mass reduction (Fig. 9H, I). To investigate the senotherapeutic effects of HHT on various cellular populations in WAT, we performed mass cytometry (CyTOF) analysis[39,67] (Fig. S15A). Intriguingly, despite no significant differences in the number of APCs in perigonadal WAT (pWAT), we found reduced expression of CD142 and ICAM-1, markers for committed APCs, in HHT-treated WAT, indicating that HHT-treated APCs maintain characteristics of stemness[39,67] (Fig. 9J, K). However, we did not find any differences in the adipose tissue immune cell populations, except for the reduction in M1-like macrophages (Fig. S15B), which is consistent with our data shown in Figs. 3E and S8E. Moreover, we did not find any differences in the histopathology of spleen tissue, blood smear, or the number of splenic lymphocytes between PBS- and HHT-treated mice, suggesting that the extended lifespan is not associated with the lymphoma suppression activity of HHT, and that HHT treatment does not cause toxicity to hematopoietic cells, liver, small intestine, or kidney (Figs. S15C–G). We found moderately reduced proteinuria in HHT-treated mice without signs of renal toxicity (Figs. S15H–M). Collectively, these results indicate that HHT delays aging and prolongs lifespan in physiological aging mice.

## Discussion

We conducted a comprehensive search among 2150 compounds currently in clinical use or undergoing phase 1 to 3 clinical trials to identify novel senotherapeutics. Our findings revealed that HHT exhibits senotherapeutic activity in vitro and in vivo. By selectively clearing senescent cells, HHT demonstrated its potential to mitigate obesity, metabolic dysfunction, and aging phenotypes, and extend the lifespan of obese and aged mice. HHT, a natural plant alkaloid derived from *Cephalotoxus hainanensis*[68], has been used for the treatment of CML in patients who are refractory or intolerant to two or more tyrosine kinase inhibitors[69]. The anti-tumor effect of HHT involves complex mechanisms, including reduced protein synthesis efficiency. Notably, HHT treatment leads to the loss of several short-lived proteins linked to cell proliferation and survival, such as c-Myc, Mcl-1, and Cyclin D1, which likely contributes to the apoptotic cell death of malignant

cells[56,70]. Additionally, HHT directly reduces the levels of anti-apoptosis genes Bcl-xL and caspase-9, further contributing to increased apoptosis of malignant cells[71]. In senescent cells, HHT augments the senolytic effect of ABT-263 via Mcl-1 inhibition[72]. These previous studies suggest that the mechanisms of HHT's pharmacological action are diverse and not yet fully understood[56,70,71]. The present study is the first to demonstrate the senotherapeutic activity of HHT, highlighting its potential for preventing and treating various age-associated diseases, including WAT aging and related metabolic complications. We further demonstrate HSPA5 inhibition as a possible mechanism for the senolytic activity of HHT.

Our findings suggest that the senotherapeutic effect of HHT is highly dependent on HSPA5, as its knockdown markedly attenuated the pro-apoptotic activity of HHT. This indicates that HHT requires HSPA5 as a functional target or signaling mediator to induce cell death, and that the physical interaction between HHT and HSPA5 may be critical for initiating apoptosis. In contrast, simple depletion of HSPA5 expression by siRNA may allow compensatory chaperone networks or stress-adaptive responses to preserve cell survival. Moreover, siRNA reduces protein levels gradually and incompletely, whereas HHT may acutely disrupt specific HSPA5 functions, including those at the endoplasmic reticulum or cell surface, that are essential for maintaining proteostasis and stress signaling. Thus, the loss of HHT-induced cytotoxicity under HSPA5 knockdown conditions supports the conclusion that HSPA5 is not only associated with but also required as a functional binding partner for HHT-mediated apoptosis, highlighting a mechanistic dependency rather than a redundant pathway.

Interestingly, animals treated with HHT exhibited reductions in body weight and fat mass without significant changes in food intake. Thus, the adipose tissue has emerged as the primary target organ when assessing the senotherapeutic and metabolic effects of HHT. WAT is highly susceptible to cellular senescence under the conditions of aging and overnutrition, contributing to metabolic abnormalities and insulin resistance. Previous studies have demonstrated that senescent adipocytes secrete SASP, which induces inflammation and promotes senescence in neighboring healthy cells[73]. Additionally, the senescence of APCs is associated with reduced subcutaneous WAT adipogenesis and adipocyte hypertrophy in humans[74].

By eliminating senescent APCs and adipocytes with HHT treatment, adipose tissue expansion can occur via hyperplasia rather than hypertrophy, reducing inflammation in the adipose tissue. In support of this, our snRNA-seq data revealed a significant increase in the number of mature adipocytes in HHT-treated WAT, indicating that HHT treatment promotes healthy adipose expansion through enhanced APC differentiation. Our CyTOF data demonstrating increased stemness of APCs following HHT treatment also supports this idea. Additionally, HHT treatment restored the acute cold-induced thermogenic activity of WAT, which is known to be reduced with aging and obesity[14,75], further demonstrating increased adipose plasticity and increased potential for metabolic benefit following HHT treatment.

The healthy remodeling of WAT was accompanied by a decrease in the number of inflammatory cells, including proinflammatory

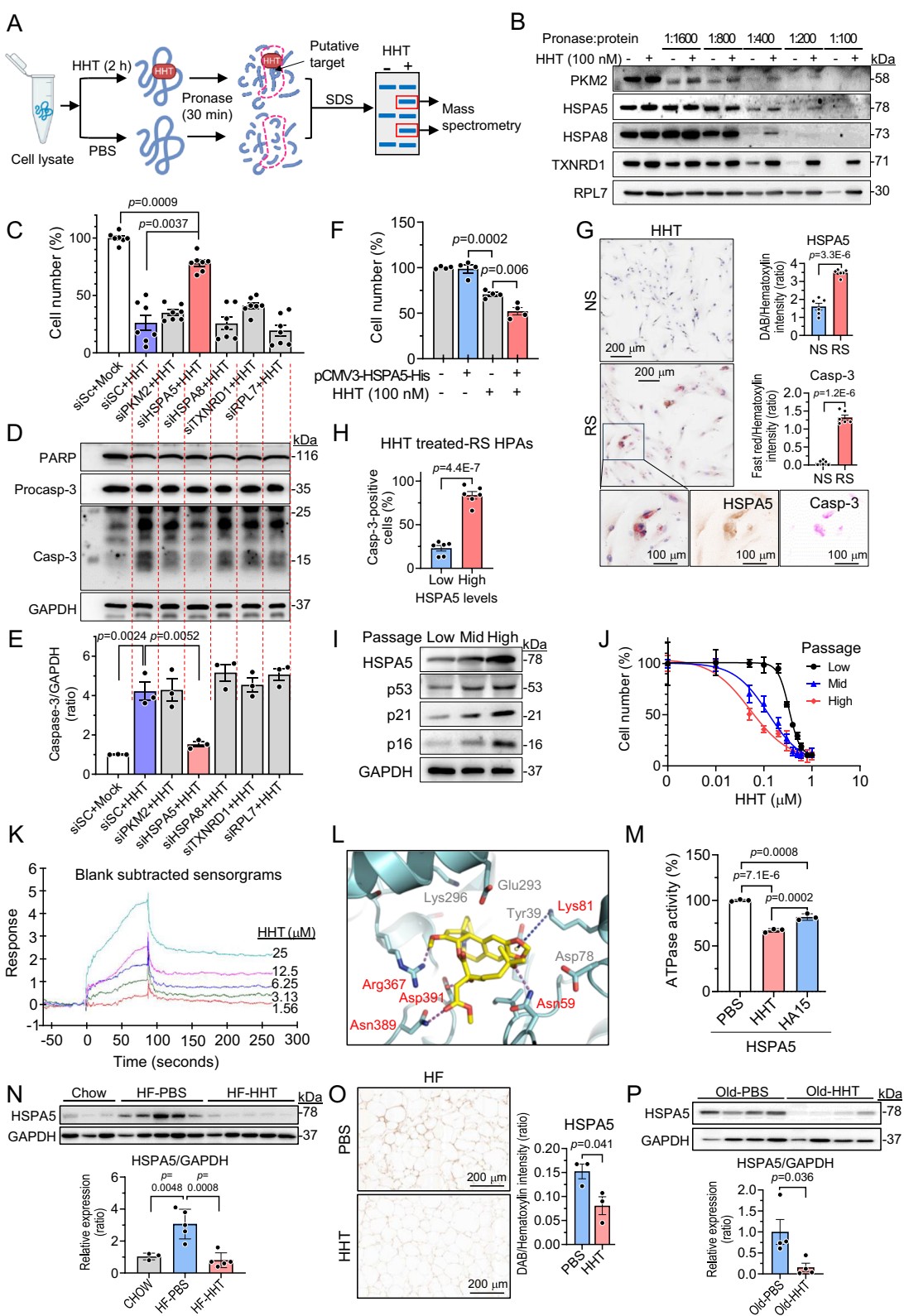

macrophages and B cells. Apoptotic cell death of hypertrophic adipocytes attracts macrophages for phagocytosis, and the accumulation of macrophage in adipose tissue induces inflammation and insulin resistance in obesity[76,77]. Thus, a reduction in macrophage number within WAT is associated with improved insulin resistance[78]. Consistently, we found that HHT treatment reduced the number of adipose macrophages, as evidenced by a decrease in crown-like structure

(CLS). However, despite the significant reduction in total macrophage numbers, HHT-treated macrophages exhibited higher levels of age-associated gene signature (e.g., SenMayo) and a stronger M1-like signature, which contrasted with the overall reduction in tissue inflammation. This discrepancy may partly reflect methodological differences between transcriptomic and protein-based analyses. Recent evidence indicates that macrophage activation states can

**Fig. 7 | HHT exerts senotherapeutic effects by targeting HSPA5. A** Schematic of the drug affinity responsive target stability (DARTS) approach. **B** Western blotting of candidate proteins in replicatively senescent human dermal fibroblasts. Replicatively senescent human visceral preadipocytes (RS HPAs) were transfected with siRNA, incubated for 4 days, then treated with homoharringtonine (HHT) or phosphate buffered saline (PBS) for 4 days (**C–E**). **C** Cell number by CCK-8 assay (n = 7 per group). **D** Immunoblots of poly (ADP-ribose) polymerase (PARP), procaspase-3 (Procasp-3) and cleaved casp-3 (Casp-3). **E** Quantification of Casp-3 (n = 3 per group). **F** Cell number by CCK-8 assay in HSPA5-overexpressing HPAs (n = 4 per group). **G** IHC images and quantification of HSPA5 (brown) and Casp-3 (red) in NS and RS HPAs after 2-day HHT treatment (n = 6 fields from 2 independent experiments). Boxed region enlarged and split into HSPA5 (brown) and Casp-3 (red) channels. **H** Quantification of Casp-3-positive cells in HHT-treated RS HPAs with high or low HSPA5 (n = 6 fields from 2 experiments). **I** Levels of HSPA5, p53, p21, and p16 proteins in HPAs at different passages. **J** HHT dose-response curves in HPAs at different passages (Low, P2–P3; Mid, P6–P7; High, P12–P13) (n = 5 per group). **K** Surface plasmon resonance affinity assay. **L** Predicted HHT–HSPA5 binding; H-bonds (magenta) and pi–cation (blue). **M** Inhibition of ATPase activity of HSPA5 by HHT and by HA15 (n = 3 per group). **N** HSPA5 levels in adipose tissues of chow diet (Ch)- and high-fat diet (HF)-fed mice administered PBS or HHT (n = 3 in Ch; n = 5 in HF). **O** IHC images and quantification of HSPA5 in adipose tissue of Ch- and HF-fed mice administered PBS or HHT (n = 3 per group). **P** HSPA5 in adipose tissues of aged mice administered PBS or HHT (n = 4 per group). Data are expressed as means ± SEM and analyzed via one-way ANOVA with a post-hoc test or two-tailed Student's t test. DAB, 3,3'-diaminobenzidine; GAPDH, glyceraldehyde 3-phosphate dehydrogenase; HSPA5, heat shock protein family A member 5; HSPA8, heat shock protein family A member 8; PKM2, pyruvate kinase M2; RPL7, ribosomal protein L7; siSC, scrambled siRNA; TXNRD1, thioredoxin reductase 1. Illustration created in part (A) with BioRender (https://BioRender.com/02ykozw).

persist even after adipose inflammation resolves, due to epigenetic imprinting or stress adaptation[79,80]. Accordingly, we propose that following HHT-induced apoptosis of senescent adipose cells, macrophages undergo transient stress responses associated with debris clearance, leading to pro-inflammatory and senescence-like features[81,82]. These findings suggest that the observed signatures represent temporary adaptive states rather than persistent inflammation, highlighting the need for future studies to define cell type–specific responses to HHT. We further speculate that these discrepancies may arise from differences in macrophage status or disease conditions; for instance, in the published study[83], HHT suppressed intestinal inflammatory signatures when administered after inflammation had already been established, whereas in our study, mice were treated with HHT during the HF feeding period. Another potential reason for the discrepancy could be the duration of treatment, as our in vivo data are based on long-term treatment, whereas the published study focuses on acute changes (9 weeks vs 1 week, respectively). Indeed, M1-like polarization of macrophages can occur within one day after the induction of adipocyte apoptosis[84]. Given that our macrophage analysis was conducted after long-term (over 9 weeks) treatment, it is likely that HHT-treated adipose tissues were undergoing tissue regeneration and remodeling rather than being in an acute pro-inflammatory state. Therefore, it seems that the healthy remodeling of WAT induced by HHT contributes to the reduction in macrophage numbers, rather than a direct anti-inflammatory effect of HHT on macrophages. Nevertheless, given these diverse effects of HHT on macrophage activation and polarization, further investigation into the effects of HHT on macrophages is warranted for future studies.

Notably, previous studies have shown that the accumulation of B cells in WAT is first detectable during middle age and is directly associated with adipose dysfunction and systemic insulin resistance[85–87]. Our data illustrate a nearly complete depletion of adipose B cells with HHT treatment. As hematopoietic suppression is a common side effect of HHT in cancer treatment[63,64], we initially suspected that the elimination of adipose B cells by HHT might be associated with its suppressive action on hematopoiesis. However, analysis of blood cell counts in mice after long-term (over 6 months) HHT treatment did not show significant differences, suggesting that depletion of adipose B cells is likely a result of HHT's senotherapeutic effect rather than its effect on hematopoiesis suppression.

Considering the systemic administration of HHT, it is anticipated that its effects may extend beyond adipose tissue and impact multiple organs. Indeed, the livers of HHT-treated animals exhibited reduced TG and ALT levels, indicating reduced hepatic steatosis and damage. In addition, HHT was found to improve fibrotic changes caused by aging in the liver, kidneys, and lungs. Among these, the changes in the lungs were significant enough to restore lung function in progeroid mice. HHT treatment also maintained the filtration function of the kidney and improved skeletal muscle function, as measured by grip strength,

in progeroid mice. One of the most notable effects of HHT treatment observed in this study was the increase in the longevity of aged mice. Importantly, this long-term treatment did not result in any adverse effects on liver and kidney function. Nonetheless, future research is needed to explore the therapeutic effect of HHT in various age-related conditions.

HSPA5, also known as glucose-regulated protein 78 (GRP78), is a molecular chaperone localized in the ER which plays an important role in protein folding and assembly. Under conditions of stress, HSPA5 is highly expressed and is translocated to the cell surface where it acts as a receptor for various endogenous and exogenous ligands[88,89]. HSPA5 is highly expressed in adipose tissue and is upregulated in patients with older age, obesity, and diabetes[90] due to increased oxidative stress, endoplasmic reticulum stress, hypoxia, and inflammation of the adipose tissue[91,92]. In addition, HSPA5 has been reported as one of the major proteins transferred by senescent cells to neighboring cells[93]. Consequently, increased expression of HSPA5 in adipose tissue is associated with adipose dysfunction. Interestingly, several anti-diabetic drugs (e.g., metformin, SGLT2 inhibitors, and β3-adrenergic receptor agonists) and non-pharmacological interventions (exercise, caloric restriction, fasting, and chronic cold exposure) have been shown to reduce the expression of HSPA5. Similarly, HHT treatment significantly reduced the expression of HSPA5 in adipose tissue. We postulate that HHT exerts metabolic benefits on adipose tissue by targeting HSPA5. This is supported by the following findings: i) upregulation of HSPA5 in senescent adipocytes and adipose tissues, ii) lack of senotherapeutic effect of HHT in senescent cells with HSPA5 knockdown, iii) enhanced senotherapeutic activity of HHT in HSPA5 overexpressed cells, and iv) direct binding of HHT to HSPA5, resulting in inhibition of its activity. Together, these findings underscore HSPA5 as a potential target of HHT, mediating its senotherapeutic action through the clearance of senescent cells in adipose tissues.

Our data clearly demonstrate that HHT treatment robustly extends the lifespan of both *Zmpste24*[-/-] progeroid and aged mice. However, given that HHT is a clinically approved drug for leukemia treatment[94,95], it is important to consider that this longevity benefit may arise not only from HHT's senolytic effect but also from its anti-leukemic effect of HHT. This dual activity raises the possibility that HHT's life-extension in our study could partly reflect protection against lethal neoplasm (e.g., lymphomas) in addition to clearance of senescent cells. Indeed, a recurring challenge in geroscience is differentiating senolytic effects from anti-cancer mechanism in longevity studies, especially since many senolytic agents are repurposed chemotherapeutics often validated in cancer-prone aged mice[96]. In this regard, our choice of a non-cancer-prone *Zmpste24*[-/-] mice minimizes confounding by spontaneous neoplasms – these progeroid mice have a severely abbreviated lifespan (4 - 5 months) and typically do not survive long enough to develop cancers[97,98], making them well-suited for isolating aging-related effects. Taken together, our findings

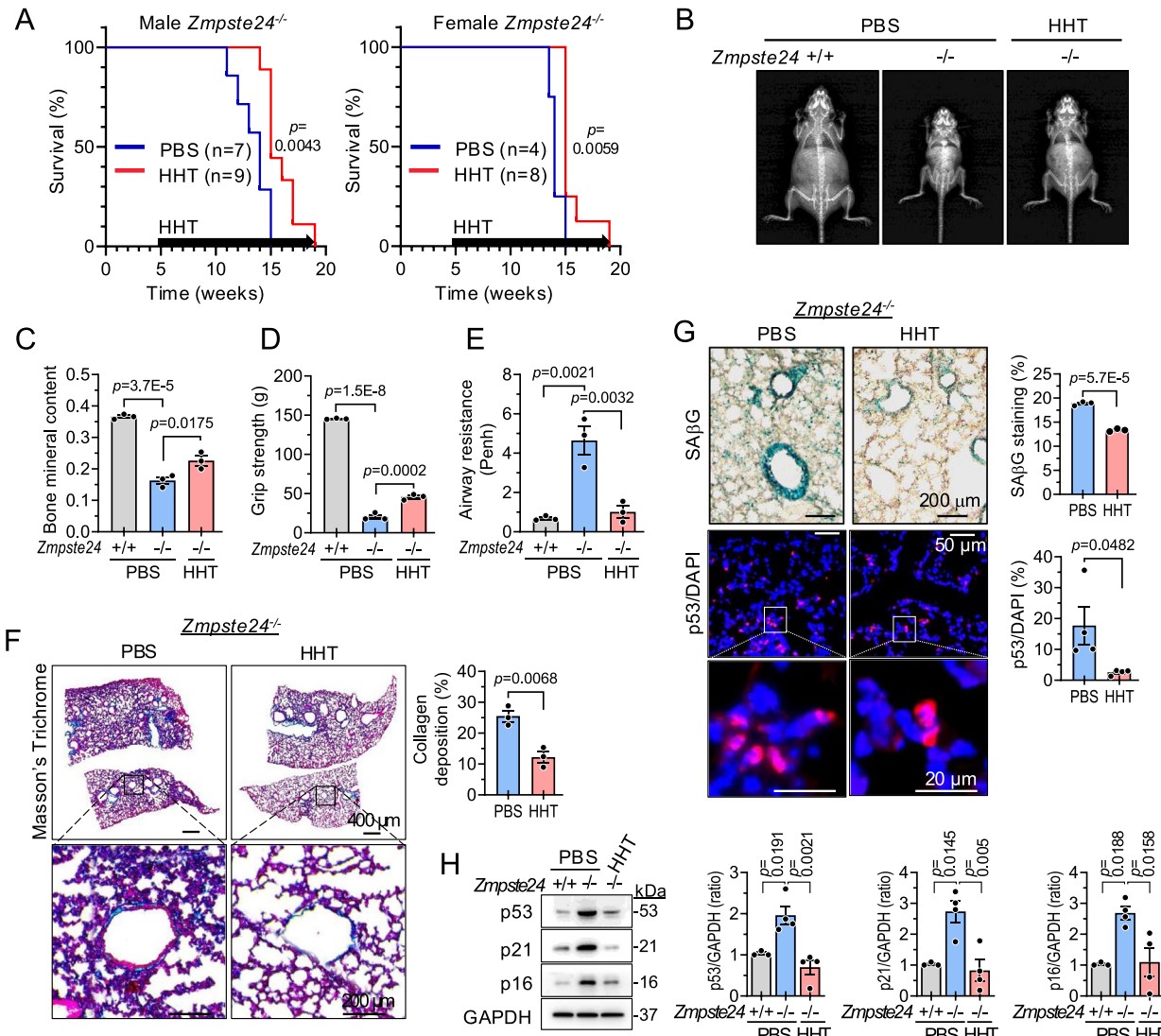

**Fig. 8 | HHT improves aging phenotypes in *Zmpste24⁻ᐟ⁻* progeroid mice.**
*Zmpste24⁻ᐟ⁻* mice (5 weeks old) were injected with HHT or PBS intraperitoneally once a week. **A** Survival curves of male and female *Zmpste24⁻ᐟ⁻* mice. The significance of the survival curves was assessed using the two-sided Gehan-Breslow-Wilcoxon test. **B**, **C** Representative DEXA images and bone mineral content of male *Zmpste24⁻ᐟ⁻* mice (*n* = 3 in each group). **D** Grip strength in male *Zmpste24⁻ᐟ⁻* mice (*n* = 3 in each group). **E** Airway resistance measured by whole-body plethysmography in male *Zmpste24⁻ᐟ⁻* mice (Penh, *n* = 3 in each group). **F** Histological analysis after Masson's trichrome (MT) staining of the lung tissues of male *Zmpste24⁻ᐟ⁻* mice (n = 3 in each group). Scale bar, 400 μm (top panel) or 200 μm (bottom panel). **G** Representative images of senescence-associated β-galactosidase (SAβG) staining and quantification (n = 3 in each group), and p53 immunofluorescence staining with quantification (*n* = 4 in each group) in lung tissues of male *Zmpste24⁻ᐟ⁻* mice. **H** Expression levels of senescence-associated proteins (p53, p21, and p16) (*n* = 3 for PBS-treated *Zmpste2⁺ᐟ⁺* mice; *n* = 4 for all other groups). Values are presented as means ± SEM. Data were analyzed via one-way ANOVA followed by a post-hoc test or via two-tailed Student's *t* test except for A. GAPDH, glyceraldehyde 3-phosphate dehydrogenase; HHT, homoharringtonine; PBS, phosphate-buffered saline.

underscore the need for further investigation to rigorously determine how much of HHT's geroprotective effect is due to senescent cell clearance versus suppression of tumorigenesis, which would be critical for guiding the future development of senotherapeutics.

In summary, our drug repositioning approach identified HHT as a novel senotherapeutic molecule. We demonstrated that HHT treatment effectively ameliorates adipose tissue dysfunction and metabolic abnormalities by targeting senescent cells in WAT. Moreover, we provided evidence of the senotherapeutic activity of HHT in human adipose tissue. Additionally, HHT treatment promoted the lifespan of aged mice without major adverse effects. Overall, our study suggests that HHT holds potential as a senotherapy for the treatment and prevention of age- and diet-induced obesity and metabolic diseases.

## Limitations of the study
It is important to acknowledge the limitations of our study. HHT-induced cytotoxicity was not observed in NS cells, particularly under serum-free (0% FBS) conditions. This resistance may be attributed to serum starvation-induced quiescence, which could enable cells to evade HHT's cytotoxic effects. Additionally, autophagy activation under serum-depleted conditions may have contributed to drug resistance, as autophagy functions as a well-established cytoprotective mechanism against various cellular stresses, including chemotherapeutic agents. We observed the most pronounced in vivo effects of HHT in adipose tissue. A plausible explanation is that HSPA5 is highly expressed in adipose tissue, particularly in visceral adipose tissue, and its expression increases with aging, obesity, and diabetes in both humans and mice[90]. These findings support the notion that adipose

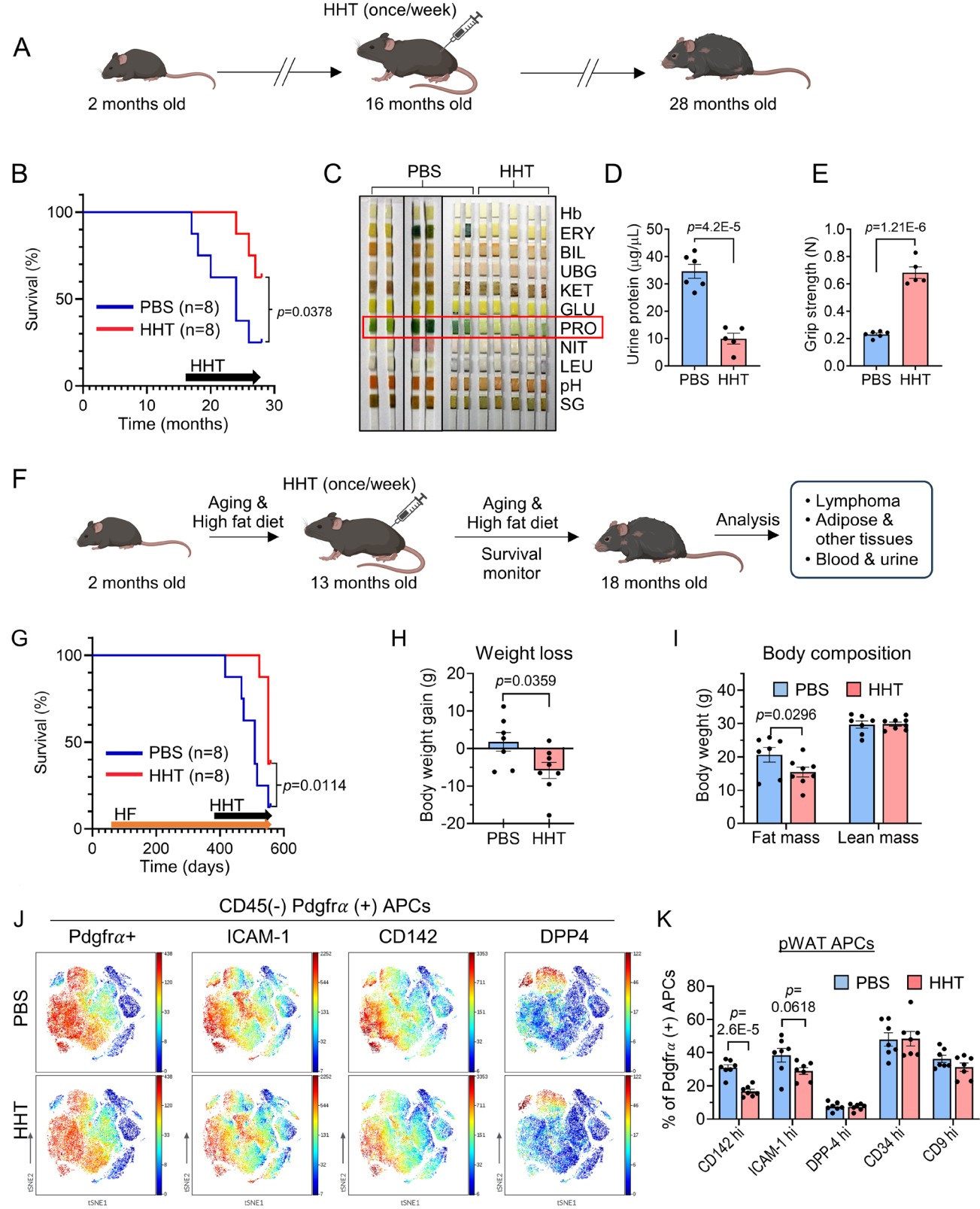

tissue may serve as a major target organ of HHT. Another possibility is that HHT preferentially accumulates in adipose tissue compared to other organs. However, tissue distribution and pharmacological action of HHT in mice are currently unavailable. Therefore, future pharmacokinetic and pharmacodynamic studies are warranted to determine the tissue distribution of HHT and to clarify its primary target sites. Since HHT was administered systemically, it may have exerted

beneficial effects on other metabolic organs/tissues, such as the liver and skeletal muscle, which could contribute to the observed weight loss and metabolic improvement. Further investigations are needed to elucidate the tissue-specific effects of HHT. Although our preclinical data and human fat explant studies demonstrate strong senotherapeutic effects of HHT, additional translational experiments are required to support future clinical trials. It would be important to

**Fig. 9 | HHT delays aging and extends lifespan in physiological aging mice.** Male C57BL/6 J mice at 16 months were injected with HHT once a week for 12 months (A-E). **A** Experimental scheme. **B** Survival curve of physiological aging mice ($n = 8$ in each group). The significance of the survival curves was assessed using the Gehan-Breslow-Wilcoxon test. **C** Urine analysis using urine strips ($n = 4$ in PBS and $n = 7$ in HHT). **D** Quantification of urine proteins ($n = 6$ in PBS and $n = 5$ in HHT). **E** Measurement of grip strength ($n = 6$ in PBS and $n = 5$ in HHT). Two-month-old male C57BL/6 J mice were fed a high-fat (HF) diet for 11 months, and HHT was administered once weekly for 5 months while continuing the HF regimen (**F–K**). **F** Experimental scheme for testing the effects of HHT on lifespan of mice. Male C57BL/6 J mice were fed HF for 11 months and injected with HHT once a week for

5 months. **G** Survival curve of HF and aging combination model. The significance of the survival curves was assessed using the Gehan-Breslow-Wilcoxon test. **H** Body weight loss ($n = 7$ in PBS and $n = 8$ in HHT). **I** Body composition ($n = 7$ in PBS and $n = 8$ in HHT). **J** CyTOF analysis of adipose precursor cells (APCs) by various APC markers. **K** Quantification of APC subpopulation in perigonadal white adipose tissues (pWAT) ($n = 7$ in each group). Values are presented as means ± SEM. Data were analyzed via two-way ANOVA followed by Sidak's multiple comparisons test or via two-tailed Student's $t$ test except for (**B**, **G**). DPP4 dipeptidyl peptidase 4; HHT homoharringtonine; hi high; ICAM-1, intercellular adhesion molecule 1; PBS phosphate-buffered saline; Pdgfrα platelet-derived growth factor receptor α. Illustration created in part (**A**, **F**) with BioRender (https://BioRender.com/y2s7wj2).

examine the effects of HHT in female mice, as our current study focused on male mice. Lastly, further research is warranted to address the potential therapeutic effects of HHT in various age-related conditions.

# Methods

## Ethical approval for animal studies and human adipose tissue research

Animal studies and human adipose tissue research were conducted in compliance with all relevant ethical regulations. Animal experiments were performed according to protocols approved by the Institutional Animal Care and Use Committee (IACUC) of Yeungnam University College of Medicine (YUMC-AEC2019-031, YUMC-AEC2020-025, and YUMC-AEC2022-039) and by the Animal Care Committee of The Centre for Phenogenomics (TCP, AUP #27-0271H). Research involving human adipose tissues was approved by the Institutional Review Board (IRB) of Yeungnam University Medical Center (YUMC 2021-03-052).

## Materials

Clinically applied compounds (2150 compounds) under phase 1 to 3 clinical trials and FDA-approved were kindly provided by Korea Chemical Bank (Daejeon, Republic of Korea) (https://chembank.org/). Cell Counting Kit-8 (CCK-8), and lactate dehydrogenase (LDH) assay kit were obtained from Dojindo Molecular Technologies (Rockville, MD, USA), 5-bromo-4-chloro-3-indolyl-β-D-galactopyranoside (X-gal) from Amresco LLC (Solon, Ohio, USA), doxorubicin from Ildong Pharmaceutical Co. Ltd. (Seoul, Republic of Korea), and HHT from Tocris (Bristol, UK). Dexamethasone (DEX), 3-isobutyl-1-methylxanthine (IBMX), and insulin were obtained from Sigma-Aldrich (St. Louis, MO, USA). Primary antibodies used in this study for Western blotting, immunostaining, and mass cytometry are listed in Supplementary Table 7 and 8. Sequences for primers used in RT-qPCR are listed in Supplementary Table 9. siRNAs were purchased from Life Technologies Corp. (Carlsbad, CA, USA), and detailed information is provided in Supplementary Table 10. HPAs (human visceral preadipocytes, #PT-5005) and HUVECs (human umbilical vein endothelial cells, #C2517A) were purchased from Lonza (Basel, Switzerland). HDFs (human dermal fibroblasts, #PCS-201-012), hRPEs (human retinal pigment epithelial cells, #CRL-2302), and 3T3-L1 mouse preadipocytes (#CL-173) were purchased from American Type Culture Collection (Manassas, VA, USA).

## Cell culture

HDFs were cultured in Dulbecco's Modified Eagle's Medium (DMEM; Welgene Inc., Gyeongsan, Republic of Korea) supplemented with 10% fetal bovine serum (FBS; Gibco, Carlsbad, CA, USA), while HPAs were maintained in a preadipocyte basal medium with appropriate supplements and growth factors (Lonza, Basel, Switzerland) containing 10% FBS. HUVECs were grown in endothelial cell basal medium-2 enriched with various growth factors, and hRPEs were cultured in DMEM:F-12 medium with 10% FBS. Additionally, 3T3-L1 cells were maintained in DMEM with 10% calf serum (Gibco).

## Preparation of prematurely senescent cells by doxorubicin treatment

To induce premature senescence (PS), HDFs and hRPEs were treated with 0.5 μM doxorubicin in serum-free media for 4 h[99]. Then, cells were cultured with DMEM (10% FBS) and DMEM/F-12 (10% FBS) for 4 days and induction of cellular senescence was confirmed by measuring the upregulation of p16 and SASP levels as well as an increase in SAβG staining. The percentages of SAβG staining in PS HDFs and PS hRPE were more than 70% and 60%, respectively.

## Preparation of replicatively senescent cells

Replicatively senescent (RS) model was established following the protocol outlined in previous studies[99,100]. Briefly, HDFs ($2 \times 10^5$), HUVECs ($2 \times 10^5$), and HPAs ($2 \times 10^5$) were cultured in a 100 mm dish and sub-cultured when cells reached 80-90% confluency. The number of population doublings (PDs) was calculated from the geometric equation: $PD = \log_2 F / \log_2 I$, where F is the cell number at the end of incubation time, and I is the cell number at the beginning of the incubation time. We also measured the population doubling time (PDT), which was calculated using the following equation: $PDT = (Tf-Ti)/\log_2(F/I)$, where Tf is the final incubation time, Ti is the initial incubation time, F is a final population number, and I is an initial population number. For all experiments, HDFs were used after passage 5 (PD < 20, PDT < 2 days) or passage 23 (PD > 60, PDT > 14 days). HUVECs were used in passage 5 (PD < 20, PDT < 2 days) or passage 23 (PD > 60, PDT > 14 days). HPAs were used in passage 3 (PD < 20, PDT < 2 days) or passage 13 (PD > 60, PDT > 14 days). These are referred to as "non-senescent (NS)" and "senescent" cells, respectively. Cellular senescence in RS HDFs, RS HPAs, and RS HUVECs was confirmed by measuring the upregulation of p53, p16, and SASP levels and an increase in SAβG staining. The percentages of SAβG staining in RS HDFs and RS HUVECs were more than 80% and RS HPAs were more than 60%.

## Cell viability assay

Cell number was assessed using the CCK-8 according to the manufacturer's instructions and by manual cell counting under light microscopy.

## Lactate dehydrogenase (LDH) assay

LDH activity was measured in media using an LDH assay kit according to the manufacturer's instructions.

## Senescence-associated β-galactosidase (SAβG) staining assay

SAβG activity was measured with minor modifications[100]. Cells were washed 2 times with phosphate-buffered saline (PBS) and fixed with 3.7% (v/v) formaldehyde in PBS for 5 min. The human adipose tissues and the epididymal WAT of mice were fixed in 4% paraformaldehyde (PFA) for 30 min and washed twice with PBS. Cells and tissues were incubated in a SAβG staining solution (1 mg/mL 5-bromo-4-chloro-3-indolyl-β-D-galactoside, 40 mM citric acid-sodium phosphate (pH 6.0), 5 mM potassium ferricyanide, 5 mM potassium ferrocyanide, 150 mM

NaCl, and 2 mM $MgCl_2$) for 18 h at 37 °C. Tissue samples were embedded with an O.C.T compound and cryosectioned to 10 µm at -30 °C. SAβG positive cells stained with blue color were counted and analyzed.

## Screening of senotherapeutic candidates from FDA-approved drugs

To identify senotherapeutic candidates from a library of 2150 FDA-approved drugs, primary and secondary screenings were conducted. In the primary screening, PS HDFs were seeded at $1 \times 10^3$ cells per well in a 96-well plate and cultured overnight. The cells were then treated with each drug at a concentration of 100 nM and incubated for 4 days at 37 °C in a $CO_2$ incubator. Cell viability was assessed using the CCK-8 assay, while cellular senescence was evaluated by SAβG staining. Based on these results, 110 compounds were selected for further analysis. In the secondary screening, RS HDFs, RS HUVECs, and PS hRPEs were seeded at $1 \times 10^3$ cells per well in a 96-well plate, cultured overnight, and treated with each drug at 100 nM for four days. Cell viability was assessed using the CCK-8 assay, and cellular senescence was determined by SAβG staining. To evaluate potential toxicity, non-senescent cells were also treated with 100 nM of each compound for four days, followed by a viability assessment using the CCK-8 assay. ABT263 and rapamycin were used as positive controls for senolytic and senomorphic activity, respectively.

## Cell cycle analysis by flow cytometry

Cells were harvested and fixed in 70% cold ethanol at 4 °C for 1 h. Cells were washed twice with PBS and resuspended in a propidium iodide (50 µg/mL, Sigma-Aldrich, P4170) solution. RNase A was added to a final concentration of 0.5 µg/mL, and the samples were incubated at 4 °C for 4 h. Flow cytometric analysis was performed using a CytoFLEX Flow Cytometer (Beckman Coulter Life Sciences, Brea, CA, USA).

## Adipocyte differentiation of 3T3-L1 cells

3T3-L1 preadipocytes were cultured in DMEM with 10% calf serum until two days post confluence (Day 0). To induce differentiation, the cells were treated with MDI differentiation medium, consisting of DMEM supplemented with 10% FBS, 0.5 mM 3-isobutyl-1-methylxanthine, 1 µM dexamethasone, and 0.5 µg/mL insulin. On Day 2, the medium was replaced with a maturation medium composed of DMEM containing 10% FBS and 0.5 µg/mL insulin. The cells were maintained in this medium for an additional 6 days, with medium changes every 2 days. On Day 8, adipocyte differentiation and cellular senescence were assessed using Oil Red O staining, SAβG staining, and Western blot analysis.

## Oil red O (ORO) staining

Cells were fixed with 4% paraformaldehyde for 30 min, rinsed 2 times with PBS, and incubated with an ORO working solution for 1 h. The images were photographed under a microscope, and ORO staining density in cells was quantified using the ImageJ software (National Institutes of Health, Bethesda, MD, USA).

## Transfection of siRNA and plasmids in HPAs

Senescent HPAs were seeded in 60 mm dishes and incubated overnight. siRNAs (20 nM) were transfected into cells using RNAiMax 2000 transfection reagent (Invitrogen, Gaithersburg, MD, USA). Protein knockdowns were confirmed by Western blot analysis. For over-expression study, HPAs were transfected with a pCMV-HSPA5-His plasmid purchased from SinoBiological (Seoul, Republic of Korea) using Lipofectamine 2000 reagent (Invitrogen) and the expression levels of HSPA5 protein were confirmed by Western blotting.

## Animals and animal experiments

C57BL/6 N mice were purchased from Koatech (Pyungtaek, Republic of Korea), and *p16-Luc* mice were kindly donated by Eiji Hara (Osaka University, Japan). *Zmpste24-/-* mice on a C57BL/6 J background were kindly gifted by Dr. Zhongjun Zhou (University of Hong Kong, China). To examine the effect of senotherapeutic candidates and HHT on adipose tissue senescence and insulin resistance in obesity, 8-week-old male C57BL/6 N mice and *p16-Luc* were randomly divided into chow diet (Envigo, Indianapolis, IN, USA) or HF groups (Research diets, New Brunswick, NJ, USA) for 6 weeks. HF contains 60.3% of the total calories as fat, 21.3% as carbohydrates, and 18.4% as protein. Then, mice in an HF group were divided into two groups, PBS- or senotherapeutic candidate-treated group, and placed on the same HF for further 6-8 weeks. Senotherapeutic candidates including HHT were dissolved in DMSO and diluted in PBS. Senotherapeutic candidates were injected intraperitoneally three times a week at the indicated dosages (Supplementary Table 2). Mice were given ad libitum access to water and food. To examine the effect of HHT on adipose tissue browning in obese mice, HF-fed mice were exposed to 4 °C for 6 h following PBS and HHT administration for 8 weeks. The HF and drinking water were provided during cold exposure.

To examine the effects of HHT on adipose tissue senescence and insulin resistance in aged mice, 18-month-old male C57BL/6 N mice (DBL, Eumseong, Republic of Korea) were randomly divided into PBS- or HHT-treated groups and were administered intraperitoneally with PBS or HHT three times a week for 14 weeks. They were provided with a chow diet.

To measure the lifespan of aged mice, C57BL/6 J (16 months old) male mice were intraperitoneally injected once a week with HHT for 12 months or PBS (n = 8 in each group). For the lifespan of *Zmpste24-/-* mice, *Zmpste24-/-* mice (5 weeks old) were administered intraperitoneally once a week for 14 weeks with HHT or PBS (n = 7 for PBS and n = 9 for HHT in male, and n = 4 for PBS and n = 8 for HHT in female mice). For the lifespan of HF-fed aged mice, C57BL/6 J (2 months old) male mice were fed an HF for 11 months, and HHT was administered once a week intraperitoneally for 5 months while continuing the HF regimen.

Mice were housed in a controlled environment with a 12 h light:12 h dark cycle. Food intake and body weight were measured manually once a week during the experimental periods. Mice were anesthetized with 2,2,2-tribromoethanol and tert-amyl alcohol (375 mg/kg), and blood was collected from the orbital plexus. The plasma and tissues were stored at −80 °C until analysis. The in vivo experiments were performed in strict accordance with guidelines and protocols approved by the Institutional Animal Care and Use Committee of Yeungnam University College of Medicine (permit number: YUMC-AEC2019-031, YUMC-AEC2020-025, and YUMC-AEC2022-039) and by the Animal Care Committee of The Centre for Phenogenomics (TCP, AUP #27-0271H).

## Ex vivo explant culture of human subcutaneous adipose tissues

This study has been approved by the IRB at the Yeungnam University Medical Center (YUMC 2021-03-052). All patients declared their informed consent in written form. Human subcutaneous adipose tissues were obtained from female patients who underwent mammoplasty after mastectomy due to breast cancer. Detailed information was provided in Supplementary Table 11. Adipose tissue samples were cut into 5 × 5 × 5 mm pieces, placed in 48 well plates, and cultivated in DMEM containing 10% FBS and antibiotics at 37 °C in a 5% $CO_2$ humidified incubator. Human subcutaneous adipose tissue samples were treated with the indicated concentrations of HHT twice every 3 days for 7 days.

## Whole-body plethysmography

Airway responsiveness was determined through whole-body plethysmography (Buxco Research System, Wilmington, NC, USA). Briefly, mice were placed in a whole-body plethysmography chamber. After a few minutes for stabilization, enhanced pause (Penh, an indication of airway resistance) was determined.

## Triglyceride (TG) measurement

The liver was homogenized in a 20-fold solution of chloroform: methanol (2:1) containing 0.01% butylated hydroxyl toluene, and vortexed for about 30 sec. After incubation overnight at 4 °C, the homogenized sample was diluted twice with distilled water and centrifuged at 800 x g for 30 min. The chloroform layer was separated, and the same amount of chloroform containing 1% triton X100 was added. After the evaporation of chloroform, the remaining part was dissolved with distilled water. Triglyceride Reagent and Free Glycerol Reagent (Sigma-Aldrich) were used to measure the level of TG. The concentration was calculated with the standard solution.

## Intraperitoneal glucose and insulin tolerance test

An intraperitoneal glucose tolerance test (IPGTT) was conducted after an overnight fasting for 16 h. Before glucose administration, blood was collected from the tail vein to measure basal blood glucose and plasma insulin levels. Animals were injected intraperitoneally with glucose (1 g/kg body weight). Blood glucose levels were measured at 0, 15, 30, 60, 90, and 120 min after injection with blood from the tail vessel, using an ACCU-CHEK Instant glucometer (Roche, Basel, Switzerland). Plasma at 0, 15, 30, and 60 min was stored at -80 °C for insulin measurement. For the insulin tolerance test (ITT), mice were fasted for 6 h and injected with insulin intraperitoneally at 0.75 IU/kg body weight. Blood glucose levels were measured at 0, 15, 30, 60, 90, and 120 min after injection using the same glucometer.

## Hyperinsulinemic-euglycemic clamp

The hyperinsulinemic-euglycemic clamp study was conducted in conscious mice 4 days after vein cannulation. The vein cannulation surgery was performed under isoflurane inhalation anesthesia, during which an indwelling catheter was inserted into the left internal jugular vein. The catheter was externalized through a small incision in the skin at the back of the neck. Following the surgery, the mice were individually housed and carefully monitored until they regained their preoperative body weight, indicating full recovery.

After an overnight fasting, hyperinsulinemic-euglycemic clamp was performed with a priming dose (900 pmol/kg body weight) of human insulin (Lilly, IN, Indianapolis, USA), and constant infusion of insulin at a rate of 24 pmol/kg/min, 20% glucose was infused for maintaining glucose at constant concentrations of 5-6 mM through a Y connector for 2 h using a microdialysis pump (CMA Microdialysis, Kista, Sweden). Blood was withdrawn from the tail vessel, and the plasma glucose levels were measured with a glucose analyzer (Analox, Middlesbrough, UK). The insulin-stimulated whole-body glucose uptake rate was estimated with a continuous injection of [3-$^3$H] glucose (PerkinElmer Life and Analytical Sciences, Shelton, CT, USA) throughout the clamps (0.1 μCi/min). The insulin-stimulated glucose uptake rate in epididymal fat was measured by 2-deoxy-D-[1-$^{14}$C] glucose (2-[$^{14}$C] DG) administration as a bolus (10 μCi) after starting the clamp at 75 min. After the clamp was over, mice were anesthetized with avertin, and collected plasma and tissue samples were stored at -80 °C for further analysis. To determine tissue 2-[$^{14}$C] DG 6-phosphate content, tissue samples were homogenized, and 2-DG-6-phosphate was separated from 2-DG in supernatants using an ion-exchange column (Bio-rad, Hercules, CA, USA). Whole body glucose turnover rate was calculated as the ratio of the glucose infusion rate to the specific activity of plasma glucose. The glucose uptake rate in each tissue was calculated from the plasma 2-[$^{14}$C] DG profile using MLAB (Civilized Software, Silver Spring, MD, USA) and the tissue 2-[$^{14}$C] DG-6-phosphate content.

## Blood and urine assay

Plasma insulin levels were measured with an enzyme-linked immunosorbent assay kit (Merck, Darmstadt, Germany). The liver enzymes, ALT and AST were measured in plasma using the commercial kit (Asan pharm, Seoul, Republic of Korea).

To collect urine, mice were held over the cling wrap and lightly stroked the belly of the animal. Urine was collected into the tube and urine protein assay was performed using a strip test (Medi-Test Combi 10 L, Allentown, PA, USA), and BCA assays (Thermo, Meridian, Rockford, IL, USA).

Urea nitrogen and creatinine in urine and serum were analyzed by using Urea Nitrogen Colorimetric Detection Kit (Arbor Assays, Ann Arbor, Michigan, USA) and QuantiChrom™ Creatinine Assay Kit (BioAssay Systems, Hayward, CA, USA), respectively according to the manufacturer's instruction. Urine total protein, albumin, creatine, potassium, and sodium were analyzed by the Beckman Coulter AU480 Clinical Chemistry Analyzer (Brea, CA, USA).

## Complete blood count and blood smear

Complete blood count analysis was performed using an automated hematology analyzer (Sysmex XN-1000 hematology analyzer, Sysmex, Kobe, Japan) in whole blood samples. Blood samples (200 μL) were collected from the orbital venous plexus of mice under isoflurane anesthesia and placed into heparinized sample tubes, and a complete blood count analysis was performed. For blood smear, following the collection of blood from mice in EDTA-containing tubes, blood smear samples were prepared on glass slides and stained using the Wright-Giemsa method[101].

## Bone density measurement with dual-energy X-ray absorptiometry (DEXA)

Body composition was assessed in *Zmpste24*$^{-/-}$ mice using a DEXA scanner (iNSiGHT VET DXA, OsteoSys Co., Ltd., Seoul, Republic of Korea). Each mouse was anesthetized with an intraperitoneal injection of 240 mg/kg avertin and placed on the scanner bed in the prone position, with the limbs and tail stretched away from the body. Quantitative data within the region of interest were obtained. Body composition of HF-fed aged mice was analyzed using EchoMRI-100 body composition analyzer (Echo Medical Systems, Texas, USA), which quantifies fat and lean mass in live, non-anesthetized mice.

## Grip strength

The forelimb grip strength of mice was measured using a BIO-GS3 grip-strength meter (BIOSEB, Pinellas Park, FL, USA). Mice were placed horizontally onto the grid (dimensions 100 × 80 mm, angled 20 °), allowed to grip the grid with their forelimbs only, and then gently pulled away backward until they released their grip. Results were averaged over 5 trials.

## Histological assessment in tissues

Tissues were immediately fixed in 10% neutral formalin. The fixed tissues were divided into an appropriate size to make a paraffin block. Blocks were cut into 5 μm using Microtome (Leica, Wetzlar, Germany). After attaching 2-3 pieces of cut sections to the glass slide, they were dried and sequentially stained with hematoxylin and eosin step. The sections attached to glass slides were first immersed in xylene for 2 min, followed by sequential rehydration in 100% ethanol for 1 minute, 95% ethanol for 2 min, and 90% ethanol for 2 min. After three brief washes in distilled water, the sections were stained with hematoxylin for 7 min, followed by three additional washes. The slides were dipped in acid-ethanol once and immediately washed. Next, the sections were incubated in sodium sulfate for 5 min and rinsed in running water.

Eosin staining was performed by immersing the sections in eosin solution for 3 min, followed by washing for 5 min under running water. The sections were then dehydrated through a graded ethanol series (70% to 100%) with three dips at each step. Finally, the sections were dipped in xylene three times and then mounted with a coverslip. The stained samples were observed under a microscope (Olympus, Tokyo, Japan) at 40× and 100× magnification. Pictures were derived using the DIXI eXcope image S/W program (DIXI Science, Daejeon, Republic of Korea), and the size of adipocytes was estimated by measuring the area of all cells in one field at 100 × magnification using the ImageJ program. Crown-like structure (CLS) percentage was analyzed by counting the number of surrounded cells by macrophages at 2 fields and then averaging them in the same samples. Paraffin-embedded tissue sections were stained with Sirius Red (Abcam, ab246832) for Picrosirius Red staining or with a Trichrome Staining Kit (Roche, 06521908001) for Masson's trichrome staining, followed by light microscopy observation. Blue collagen-positive areas in Masson's trichome staining were analyzed using an ImageJ software.

## Immunofluorescence staining

For γH2AX, F4/80, and p53 staining, paraffin-embedded tissue sections were deparaffinized and subjected to heat-induced epitope retrieval in citrate buffer (pH 6.0) using a pressure cooker. Sections were then incubated with Protein Block (Agilent, X090930-2) or 5% BSA (MP Biomedicals, 160069) in PBS to reduce non-specific binding, followed by overnight incubation at 4 °C with primary antibodies diluted in DAKO Antibody Diluent (Agilent, S302283-2). Rabbit anti-γH2AX (Fortis Life Science, IHC-00059; 1:1500), rabbit anti-iNOS (Abcam, #ab3523; 1:100), and rabbit anti-p53 (Abbkine, ABP0110; 1:200) were used. Corresponding secondary antibodies - donkey anti-rabbit IgG Alexa Fluor 647 (Invitrogen, A32795), goat anti-rat IgG Alexa Fluor 647 (Invitrogen, A48265), and goat anti-rabbit IgG Alexa Fluor 546 (Invitrogen, A11035) - were applied at room temperature. Nuclear counterstaining was performed with 4′,6-diamidino-2-phenylindole (DAPI) (Sigma-Aldrich, D9542). To reduce tissue autofluorescence, slides were incubated in Sudan Black B (Sigma-Aldrich, 199664) saturated in 70% ethanol for 25 min at room temperature, then mounted with Vectashield Vibrance anti-fade medium (Vector, H-1700). Whole-slide images were acquired at 20× magnification using an Olympus VS120 scanner equipped with a Hamamatsu ORCA-R2 C10600 digital camera (Evident, Japan).

## Immunohistochemistry

Paraffin-embedded tissue sections were deparaffinized, treated with freshly prepared 0.45% $H_2O_2$ in methanol for 15 min at room temperature in the dark, and subjected to heat-induced epitope retrieval in 10 mM citrate buffer (pH 6.0) at 95–98 °C for 15 min. Cell samples were fixed with 4% paraformaldehyde for 15 min and treated with 3% $H_2O_2$ for 10 min. Both tissue sections and cell samples were permeabilized with 0.2% Triton X-100 in TBS (tris-buffered saline) for 10 min. Tissue sections were blocked with 5% normal donkey serum containing 1% bovine serum albumin (BSA), and cell samples were blocked with 5% BSA (MP Biomedicals, 160069) in PBS for 1 h. Tissue sections were then incubated with a mouse anti-HSPA5 antibody (Santa Cruz Biotechnology, sc-166490; 1:100) at 4 °C overnight. Cell samples were incubated with a mixture of a mouse anti-HSPA5 antibody (Santa Cruz Biotechnology, sc-166490; 1:100) and a rabbit anti−cleaved caspase-3 antibody (Cell Signaling Technology, #9664S; 1:100) at 4 °C overnight. After three washes with TBS-T (TBS containing 0.05% Tween-20), secondary antibodies were applied for 1 h at room temperature. For tissue sections, a horse radish peroxidase (HRP)-conjugated goat anti-mouse secondary antibody (Cell Signaling Technology, #7076) was used. For cell samples, a mixture of HRP-conjugated goat anti-mouse antibody (Invitrogen, #31430; 1:50) and alkaline phosphatase (AP)-conjugated goat anti-rabbit antibody (Sigma-Aldrich, A3687; 1:50) was

applied. Following final TBS-T washes, color development was performed using enzyme-specific substrate systems: HRP activity was visualized with 3,3′-diaminobenzidine (DAB) substrate (Cell Signaling Technology, #7076, or Dako, K5007) to yield brown staining, and AP activity was visualized with Fast Red TR/Naphthol AS-MX (Sigma-Aldrich, F4648) to yield red staining. Finally, tissue sections and cell samples were counterstained with hematoxylin and mounted. Images were acquired using a KF-PRO-005 slide scanner (KFBIO Technology for Health, Yuyao, China), and quantification of stained areas or positive cells was performed using ImageJ software (National Institutes of Health).

## Metabolic cage analysis

Metabolic assessments of each mouse were conducted for a continuous 48 h period using individual closed metabolic chambers (Oxylet; Panlab, Cornellà, Spain). During the measurement period, all mice were provided ad libitum access to food and water to maintain normal physiological conditions. Oxygen consumption ($VO_2$, mL/min) and carbon dioxide production ($VCO_2$, mL/min) were sequentially measured for 3 min per mouse, rotating through all animals housed in the system. Room air was analyzed for 3 min after completing one cycle of all mice to establish baseline gas composition. All collected data, including $VO_2$, $VCO_2$, and energy expenditure, were analyzed using the manufacturer-provided software (Metabolism; Panlab). To account for inter-individual differences, all values were normalized to body weight.

## Epigenetic age analysis

The genomic DNA of adipose tissue was prepared using a NucleoSpin DNA Lipid Tissue Kit (Machery-Nagel, Duren, Germany), and the identification of differentially methylated CpG (DMCs) using reduced representation bisulfite sequencing (RRBS) data with 500 ng of genomic DNA.

Briefly, the genomic DNA was digested with MspI and ApeKI and then purified using the Qiagen MiniElute PCR Purification Kit. The DNA fragments were directly subject to end-blunting followed by adding a single adenine nucleotide (dA-tailing) to prepare the DNA for adapter ligation. Methylated adapters were ligated to the prepared DNA, and the adapter-ligated fragments, specifically those within the size range of 160 to 420 bp, were isolated and used for bisulfite conversion utilizing the ZYMO EZ DNA Methylation-Gold Kit™, adhering to the manufacturer's guidelines. The conversion process alters unmethylated cytosines to uracils, enabling subsequent methylation analysis. Following bisulfite treatment, the libraries were amplified via PCR using PfuTurbo Cx Hotstart DNA polymerase (Agilent Technologies, Santa Clara, CA, USA), a step that enriches the library for subsequent sequencing. The final RRBS libraries were evaluated for quality and size distribution using an Agilent 2100 Bioanalyzer (Agilent Technologies). The RRBS libraries were sequenced on an Illumina NextSeq 500 (NextSeq 500, Illumina, San Diego, CA, USA).

Sequencing adapters and low-quality bases present in the raw reads were trimmed using the Skewer tool[102]. Following the quality control and adapter trimming, the resulting high-quality reads were aligned to a reference genome that had been converted to a 3-letter code (to represent bisulfite-treated DNA) using the bs_seeker2-align module of BSseeker2[103]. After alignment, the bs_seeker2-call_methylation module of BSseeker2 was employed to determine the methylation status at CpG sites across the genome based on the mapped reads with a minimum coverage of 5 to obtain a detailed profile of methylation patterns, enabling the identification of methylation levels at individual CpG sites across the samples in control and treatment groups.

To estimate epigenetic age (DNAm age), we employed a multi-tissue epigenetic age clock model that utilizes ridge regression model for parameter estimation, leveraging DNA methylation data across

various tissues. The parameters for this model, including beta coefficients (weights) and intercepts for methylation CpG sites, were sourced from supplemental information reported in the manuscript[104]. These parameters are well-validated and have demonstrated robustness in predicting chronological age from methylation data across different tissues, including adipose tissue. For each sample in our dataset, epigenetic age was calculated by summing the products of methylation levels at specific CpG sites and their corresponding beta coefficients. Recognizing the challenges associated with the variable number of clock sites available in RRBS, and to enhance the accuracy and consistency of our age estimates, we imputed missing methylation data by filling zero coverage sites with the mean methylation level observed for those sites within the control or treatment groups[105]. This method ensures a more robust estimation of DNAm age by maintaining a comprehensive dataset, thereby reducing the bias from missing data. The resulting calculation provides a weighted sum of methylation levels, representing an estimation of the epigenetic age in months.

## Protein extraction and Western blotting

Cells were washed with ice-cold PBS and harvested by scraping in 50 µl of ice-cold radioimmunoprecipitation assay (RIPA) buffer (25 mM Tris-HCl, pH 7.4, 150 mM KCl, 5 mM NaF, and 1 mM phenylmethylsulfonyl fluoride). Cells were ruptured by vortexing two times for 30 sec at 30 min intervals on ice and centrifuged at 16,000 x g for 15 min at 4 °C. To measure SASP in culture media, the culture media were filtered with an Amicon® Ultra-4 Centrifugal Filter, 50 kDa MWCO (Merck KGaA, Darmstadt, Germany) and concentrated using an Amicon® Ultra-4 Centrifugal Filter, 3 kDa MWCO (Merck KGaA). Human adipose tissues were homogenized in 5 volumes of ice-cold RIPA buffer and centrifuged at 16,000 x g for 15 min at 4 °C. Protein concentrations in the supernatants were quantified by the bicinchoninic acid method (Pierce Biotechnology Inc., Rockford, IL, USA). Mouse adipose and lung tissue samples were homogenized in 10 volumes of ice-cold lysis buffer containing 50 mM HEPES, 150 mM NaCl, 50 mM NaF, 1 mM phenylmethylsulfonyl fluoride (PMSF), 1 mM benzamide, 1 mM EDTA, 1 mM EGTA, 1 mM sodium ortho-vanadate, 0.22% β-glycerophosphate, 1% NP40, 10% glycerol, and protease inhibitor cocktail (Santa Cruz Biotechnology., Santa Cruz, CA, USA). The tissue homogenate was centrifuged at 16,000 x g for 10 min at 4 °C, and the supernatant containing protein was collected. Protein concentrations were determined using the Bradford (Bio-rad) assay. An equal amount of protein samples was separated on 10% sodium dodecyl sulfate (SDS)-polyacrylamide gel electrophoresis at 70 V, and then resolved proteins were transferred to a membrane. Membranes were blocked in 1 × TTBS (10 mM Tris-HCl pH 7.5, 150 mM NaCl, and 0.05% Tween-20) containing 5% skim milk for 30 min at room temperature. Primary antibodies were applied overnight at 4 °C, and then HRP-conjugated secondary antibodies were applied for 90 min at room temperature. After rinsing the membranes 3 times with 1 × TTBS for 30 min, antigen-antibody complex was detected using Luminol Reagent (Santa Cruz Biotech Inc). Proteins were visualized with the Fujifilm LAS-3000 image system (Stamford, CT, USA). Glyceraldehyde 3-phosphate dehydrogenase (GAPDH) was used as a control for protein loading. The relative intensities of protein bands, as compared with that of the respective GAPDH signal, were determined by using the Multi Gauge software, version 3.0 (Fujifilm Corp., Stamford, CT, USA), and normalized to 1.0 for control cells by averaging three separate experiments. All phosphorylated forms were normalized to their respective protein forms.

## Reverse transcription quantitative real-time polymerase chain reaction (RT-qPCR)

Total RNA was extracted from 80 mg tissue samples using RNeasy Lipid Tissue Mini Kit (QIAGen, Hilden, Germany) according to the manufacturer's instructions. cDNA was synthesized from 1 µg of RNA using a reverse transcription kit (Applied Biosystems, Foster City, CA, USA). RT-qPCR was performed using the Real-time PCR 7500 System and Power SYBR Green PCR Master Mix (Applied Biosystems). The expression levels of GAPDH were used for sample normalization. Each reaction mixture was incubated at 95 °C for 10 min followed by 45 cycles of 95 °C for 15 sec, 55 °C for 20 sec, and 72 °C for 35 sec.

## Single nucleus RNA sequencing and data analysis

For adipose tissue single nuclei sequencing (snRNA-seq), frozen tissues were chopped into smaller pieces on dry ice. Chopped tissues were immediately incubated in lysis buffer before homogenization in glass douncers. Isolated nuclei were then washed twice with cold wash and resuspension buffer. Nuclei suspension was filtered through 40 µm filter before proceeding to the next step. Nuclei were stained with DAPI and flow-sorted. After sorting, nuclei were washed and resuspended for counting. Nuclei were stained with SYBR Green II and counted under the microscope. Sorted nuclei were used as input into the 10x Genomics single-cell 3' v3.1 assay (Pleasanton, CA, USA) and processed as described by the protocol provided by the 10x Genomics. For sequencing, the molarity of each library was calculated based on library size as measured by bioanalyzer (Agilent Technologies) and qPCR amplification data (Roche). Samples were pooled and normalized to 1.5 nM. The library pool was denatured using 0.2 N NaOH for 8 min at room temperature and neutralized with 400 mM Tris-HCl. Library pool at a final concentration of 300 pM was loaded to sequence on Novaseq 6000 (Illumina). Samples were sequenced with the following run parameters: Read 1–28 cycles, Read 2–90 cycles, index 1–10 cycles, index 2–10 cycles. Sequencing target reads for each library were 40,000–50,000 reads/nuclei.

The 10x Genomics' CellRanger v5.0.0 software was used for sequence alignment and barcode processing with the mouse reference genome (mm10-2020-A). The filtered gene-count matrices from CellRanger were used for subsequent preprocessing. Doublets were identified and removed through the automated 'scDblFinder' toolkit using default parameters[106]. Low-quality nuclei were removed from the dataset by retaining only nuclei with greater than 400 unique features (genes), less than the 95th percentile of total UMI counts & features, and less than 0.5 percent of mitochondrial transcripts detected. The percentage of measured gene expression in each cluster attributed to mitochondrial gene (percent. MT) and the number of unique genes found for each nucleus (nFeature_RNA) were within expected values for each cell[107].

The filtered data from HHT treated and control sample was analyzed using the standard integration protocol as implemented in Seurat v4.0.1[108,109]. This involved log normalization, variable feature selection and selecting/finding integration features/anchors. Cells were clustered using unsupervised Louvain clustering using the integrated nearest neighbors graph as input. The clustering resolution was determined empirically, as defined by the resolution with the greatest cluster stability (sc3_stability) from the clustree R package[110,111]. In the final dataset, there were 5342 and 6392 cells in the HHT and control conditions, with 14 total clusters, each present in both conditions.

Cluster-specific markers were identified using the Wilcoxon Rank Sum test implemented in the *FindAllMarkers* function in Seurat. Lastly, differential gene expression between HHT and control was modeled using a hurdle MAST model[112]. We considered genes with an absolute fold change greater than 0.10 and Bonferroni-adjusted $p$ value < 0.05 to be differentially expressed between conditions. These differentially expressed genes were then used as input for Gene Ontology (GO) enrichment analyses. Specifically, GO analysis was conducted for biological processes ontology with a minimum gene set size of 10 and maximum size of 500 using the clusterProfiler package[113]. For age-related analysis on snRNA-seq data, we performed additional analysis using SenMayo signature. SenMayo scores were calculated using Seurat's 'AddModuleScore' function[114], which computes the average expression levels of each gene set− in this case, the SenMayo set[46]−at

the single-cell level. These scores are then normalized by subtracting an aggregated expression of control feature sets. To account for differences in total expression, all analyzed genes were first binned based on their average expression, and control features were then randomly selected from each bin. The SenMayo signature scores were subsequently compared between PBS- and HHT- treated groups using Wilcoxon's rank sum test, with $p$ values adjusted for multiple comparisons using FDR correction.

### Drug affinity responsive target stability (DARTS)
The DARTS procedure was performed as previously described[51]. Briefly, cell lysates of RS HDFs were incubated with 50 μM HHT for 2 h at room temperature with rotation, and then 2 μl of the indicated protease solution was added to 20 μL samples of lysate. After incubation for 30 min at room temperature, digestion was stopped by adding a protease inhibitor cocktail. Proteins were separated by SDS-PAGE and observed with silver staining. Gel bands showing increased stability were cut out and prepared for mass spectrometry analysis to characterize proteins (Genomine Biotechnology, Pohang, Republic of Korea).

### ATPase activity measurement
ATPase activity was measured by an enzyme-coupled spectrophotometric assay using ATPase Activity Assay Kit (Sigma-Aldrich, #MAK113) following the manufacturer's instructions. Briefly, we established a standard curve using phosphate standards (#MAK113B). The reaction solution (40 mM Tris, 80 mM NaCl, 8 mM $MgAc_2$, 1 mM EDTA (pH 7.5), and 4 mM ATP) was incubated with HSPA5 protein (5 μg) alone or HSPA5 protein and 10 μM inhibitor (HA15) at room temperature for 60 min. Each well was added with 200 μl of reagent (#MAK113A) and then incubated for 30 min at room temperature to terminate the enzyme reaction. The optical density of colorimetric product was measured using a microplate reader (TECAN, Männedorf, Switzerland) at 620 nm. The obtained absorbance was substituted into the standard curve to calculate the ATPase activity.

### Surface plasmon resonance
Measurements of the apparent dissociation constants (KD) between HSPA5 and HHT were carried out using a Biacore T200 biosensor (GE Healthcare Bio-Sciences, Pittsburgh, PA, USA). The recombinant human HSPA5 was bound covalently to the sensor chip CM5 (carboxylated dextran matrix) using an amine-coupling method. The human HSPA5 (20 μg/mL) in 10 mM sodium acetate pH 4.0 was coupled via injection, followed by the injection of 1 M ethanolamine to deactivate residual amines. For kinetic measurements at 20 °C, HHT with concentrations ranging from 1.5625 to 100 μM was prepared by dilution in HBS-P buffer (10 mM of HEPES (pH 7.4), 150 mM NaCl and 0.05% v/v surfactant P20 and 2% DMSO). Based on the results, the association rate constants (ka), dissociation rate constants (kd), and the equilibrium dissociation (binding) constants (KD; kd/ka) were determined.

### Docking study
Possible binding model between HSPA5 and HHT was predicted by docking studies based on the X-ray structures of HSPA5 and its known inhibitor VER155008. The binding site of HHT is presumed to be like VER155008 and the related PDB structure (6CZ1.pdb) was referenced for docking study. The predicted binding model was verified considering the interaction between HSPA5 and VER155008. The initial binding model was determined by rigid docking study and MM-GBSA was performed for the final binding model prediction. MAESTRO (Schrödinger LLC, NY, USA) was used all calculational processes using Glide and Prime modules on Linux environment with default parameters[115].

### Mass cytometry (CyTOF) study and data processing
For CyTOF, stromal vascular fraction (SVF) and splenic immune cells were isolated from WAT and spleen, respectively. Visceral WAT and spleen are dissected, and finely minced. Adipose tissue was digested with Adipose tissue dissociation kit (Miltenyi Biotec, Gaithersburg, MD, USA) using gentle MACS™ Octo Dissociator with Heater (Miltenyi Biotec, Gaithersburg, MD, USA). Splenic cells were washed with PBS, neutralized with DMEM/F12 media supplemented with 10% FBS, and filtered using a 100 μm filter. WAT tissues were further digested with Type II collagenase (Worthington; #LS004176) in a 37 °C incubator for 30-40 min. Digested tissue was neutralized with DMEM/F12 media supplemented with 10% FBS and filtered using a 100 μm filter. The flow-through was centrifuged for 5 min at 400 x g. After removing the floating adipocyte fraction and supernatant, red blood cell lysis buffer (Sigma-Aldrich, #R7757) was added and incubated at room temperature for 2-3 min. The red blood cell lysis buffer is neutralized with DMEM/F12 media (supplemented with 10% FBS) and centrifuged for 5 min at 400 x g. The dissociated cells from adipose tissue or spleen were washed in a cell staining medium (CSM) and each sample was barcoded before staining according to the protocol described in the Cell-ID 20-Plex Pd Barcoding kit (Fluidigm, San Francisco, CA). Barcoded samples were pooled into 1 sample and stained with metal-tagged antibodies (Supplementary Table 8) as previously described[67]. Fixation and permeabilization were conducted according to the manufacturer's instructions (Foxp3/Transcription Factor Staining Buffer Set, eBioscience). Finally, samples were washed and suspended in Maxpar Cell Acquisition Solution containing EQ normalization beads (Fluidigm, #201237) and data was collected by Helios instrument. FCS files were normalized using EQ beads by the Helios software. All data processing and analysis was done with Cytobank Enterprise (Beckman Coulter).

### Statistical analysis
All statistical analyses were assessed by using Prism version 8 (Graph Pad Software Inc; La Jolla, CA, USA). The results are presented as the means ± standard error. In in vitro experiments, we conducted at least three independent experiments. Differences among the three or more groups were analyzed via one-way or two-way analysis of variance (ANOVA) followed by Tukey HSD test. Differences between two groups were analyzed by two-tailed Student's $t$ test. A $p$-value below 0.05 was considered statistically significant.

### Reporting summary
Further information on research design is available in the Nature Portfolio Reporting Summary linked to this article.

## Data availability
Adipose tissue snRNA-seq data are available from the National Center for Biotechnology Information (NCBI) under accession number PRJNA1260162. The main data are available within this article and its Supplementary information/Source data file. Source data are provided with this paper.

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

## Acknowledgements

S.-Y.P. and J.-R.K. are supported by the Medical Research Center Program (2022R1A5A2018865 and 2015R1A5A2009124) and the Basic Science Research Program (S.-Y.P., 2019R1A2C1088730; E.-C.K., 2021R1I1A1A01056317; J.-R.K., 2022R1A2C2004099) through the National Research Foundation of Korea (NRF) funded by the Ministry of Science and ICT. H.-K.S. is supported by grants from Canadian Institute of Health Research (CIHR, PJT-162083, 190016), Natural Sciences and Engineering Research Council (NSERC, RGPIN-2016-06610) of Canada, Diabetes Canada (OG-3-23-5715-HS), Canada Foundation for Innovation (CFI, #40249), and Sun Life Financial New Investigator Award of Banting & Best Diabetes Centre (BBDC) of University of Toronto. J.H.L. is supported by Doctoral Program Postgraduate Scholarship (PGS-D) from Natural Sciences and Engineering Research (NSERC).

## Author contributions

E.-C.K., H.-B.J., H.-K.S., J.-R.K., and S.Y.P. conceived, designed, and performed the research. E.-C.K. and H.B.J. performed in vivo and in vitro experiments. H.-N.C., M.C., S.P., Y.-K.P., M.-G.S. and Y.S. assisted in vitro and in vivo experiments. Y.P., J.H.L., Q.Z., and J.Y. performed single-nucleus RNA sequencing analysis and fibrosis staining of adipose tissue. L.P. analyzed adipocyte size. I.-K.K. provided human subcutaneous adipose tissues. B.R.O., N.V., A.Y., Q.Z., J.T., S.-R.J., H.-J.Y., and K.N.E. assisted in vitro and in vivo experiment. S. M. analyzed the epigenetic age of adipose tissue. S.L., Y.H., and J.-Y.L. performed the docking study. H.-K.S., J.-R.K., and S.Y.P. analyzed data and wrote the manuscript. All authors discussed the results and agreed to the final version of the manuscript.

## Competing interests

The authors declare no competing interests.

## Additional information

[1]Senotherapy-based Metabolic Disease Control Research Center, College of Medicine, Yeungnam University , Daegu, Republic of Korea. [2]Department of Biochemistry and Molecular Biology, College of Medicine, Yeungnam Unniversity, Daegu, Republic of Korea. [3]Department of Physiology, College of Medicine, Yeungnam University, Daegu, Republic of Korea. [4]Translational Medicine Program, The Hospital for Sick Children, Toronto, ON, Canada. [5]Department of Laboratory Medicine and Pathobiology, University of Toronto, Toronto, ON, Canada. [6]Department of Plastic and Reconstructive Surgery, College of Medicine, Yeungnam University, Daegu, Republic of Korea. [7]AI Molecular Design Team, New Drug Development Center, Daegu Gyeongbuk Medical Innovation Foundation (K-MEDI hub), Daegu, Republic of Korea. [8]These authors contributed equally: Eok-Cheon Kim, Han-Byul Jung.
✉e-mail: hoon-ki.sung@sickkids.ca; kimjr@ynu.ac.kr; sypark@med.yu.ac.kr

