## [Transparent Peer Review file · Nature Communications]

Homoharringtonine exhibits senotherapeutic activity that mitigates diet- and age-associated obesity and insulin resistance and extends lifespan in mice

Corresponding Author: Dr Hoon-Ki Sung

Version 0:

Reviewer comments:

Reviewer #1

(Remarks to the Author)

The authors have made a commendable effort in addressing the previous reviewer concerns and added substantial new data. I have a few more comments which, if addressed, would further improve the manuscript for publication.

The study relies primarily on LDH and CCK-8 assays to test cytotoxicity of senescent cells. However, these methods are based on cellular metabolism, and senescent cells are known to have abnormal metabolic profiles that can skew such readouts. Please validate the cell viability/cytotoxicity data using metabolism-independent methods, such as nuclear staining for direct cell counting (DAPI/Hoechst) to corroborate findings.

Common senescence markers of p16 and p21 are not interchangeable in many cases as they may represent distinct senescent subtypes and have different biological consequences. The authors used only p16 to evaluate senescence in vitro while they focused on p21 analysis in vivo tissues. It is unclear whether HHT targets both populations equally or selectively. It would be helpful to provide data on both p16 and p21 expression in the models studied.

The results on macrophage M1/M2-like signatures appear inconsistent. While the authors provide some reasonable interpretations in the manuscript, another possibility is that a subset of macrophages may themselves become senescent, particularly in aged or metabolically stressed adipose tissue, and contribute to the confounding transcriptomic signals. It would be worthwhile to test whether senescent macrophages are present in these tissues and whether HHT treatment impacts this population.

Under 0% FBS conditions, cells may enter quiescence. Please check quiescent markers (e.g., p27KIP1, low Ki-67) to see if the control cells are simply quiescent as it would confound the interpretation of "selective killing".

The strongest in vivo effects of HHT are seen in adipose tissue. Is this because HHT accumulates preferentially in fat, or that its target HSPA5 is expressed in adipose to a greater extent than other tissues? Biodistribution or PK/PD data would be informative.

To further support HSPA5 as the senotherapeutic target of HHT, the authors should provide data showing HSPA5 expression in senescent vs non-senescent cells, and across the major tissues affected. This would better support the proposed mechanism of action.

HHT extends lifespan in *Zmpste24*^{-/-} and aged mice - median lifespan, maximum lifespan, or both? And how much % increase?

Minor point. Please define abbreviations (Ep, RP, SC, and BAT) in Figure 2C and the main text where they first appear. And please provide the PCR primer sequences used in the method section.

Reviewer #2

(Remarks to the Author)

In the revised manuscript "Homoharringtonine, a novel senotherapeutic, mitigates diet- and age-associated obesity and insulin resistance and extends the lifespan of mice", Kim et al have indeed addressed the insufficient description of their experimental methods. However, in terms of experimental content, they still fail to convince me that they truly understand the molecular mechanism of how HHT eliminates senescent cells. Although the effects of drug treatment seem to be clear in animal experiments, the data and arguments are inconsistent and logically inadequate when authors investigate the impact of HHT on the immune microenvironment. These inconsistencies may suggest either instability in data quality or that the authors' hypothesis regarding the senolytic effect of HHT is not entirely valid, thereby diminishing the clinical relevance of the study. However, if the focus is solely on HHT's effects in a high-fat diet model, I believe the authors have made a considerable effort in this aspect.

1. Regarding the authors' response to Point 2, I now understand the reason for using 0% FBS. In both studies of ABT263 and D+Q, apoptosis is a critical indicator for evaluating senolytic efficacy, rather than changes in cell number. Therefore, in conclusion, I believe the authors are not required to perform experiments under 0% FBS conditions (for reasons to be explained below). However, they do need to clearly demonstrate whether the reduction in cell numbers for senescent and normal cells is due to cell cycle arrest or apoptosis. However, since 100 nM HHT reduced the number of NS HPAs and HDFs by approximately 40% and 25%, respectively, it is essential for the authors to describe how HHT passed the "toxicity to NS cells" screening threshold in Fig. S11.

As the authors noted, low concentrations of FBS reduce cell proliferation, inducing a quiescent state and potentially promoting autophagy. While the authors intended to demonstrate that HHT was not toxic to normal cells undergoing cell cycle arrest, the observed difference could also result from autophagy-mediated drug resistance. Moreover, it remains unclear how HSPA5 expression and translational activity are affected under these conditions, which further confounds the explanation of the differential response between senescent and normal cells.

2. As the authors mentioned, if HHT reduces p53 and p21 levels by inhibiting translation, this mechanism does not explain how HHT induces G1 cell cycle arrest in normal cells. Furthermore, it also fails to illustrate why p53 and p21 levels remain unchanged in 0% FBS-treated NS cells, as shown in Fig. 1K. In addition, the description of Fig. 1K was missing in the manuscript.

3. Regarding the authors' response to Point 6, they cited several references to argue that changes in the M1/M2 ratio within adipose tissue may not reliably reflect the inflammatory state. Through this reasoning, they attempt to rationalize the inconsistency between the increased M1/M2 ratio observed in the snRNA-seq data and the results from histological analyses. However, among the cited literature, one study explicitly states that although classical markers may not clearly distinguish M1 and M2 macrophages, the MacSpectrum characterization can still identify M1-like and M2-like phenotypes (PMID: 30990466). This study also confirmed that in obese mice, M1-like macrophages increase while M2-like macrophages decrease.

If the authors insist that neither classical markers nor transcriptomic signatures can indicate the inflammatory status of macrophages in adipose tissue, then why do they present a separate analysis of M1 and M2 populations in the CyTOF data shown in Fig. S15B, claiming a significant decrease in M1 macrophages? Is it possible that the authors separated M1 and M2 populations in the analysis because combining them would not have resulted in statistical significance?

In addition, the authors stated that "to precisely assess the inflammatory status of ATMs, they analyzed CLS using F4/80 staining." However, as the authors mentioned, F4/80 is a pan-macrophage marker expressed by various macrophage subtypes. In fact, if expression level is to be considered, pro-inflammatory macrophages—such as M1 or certain monocyte-derived populations—may express even lower levels of F4/80 compared to M2 or tissue-resident macrophages, as noted in the references cited by the authors and others (PMID: 37651193). Furthermore, according to the CyTOF result, M1 macrophages constituted only a small fraction of the total macrophage population, which was strongly inconsistent with the F4/80 staining patterns observed in CLS. Because both CyTOF and snRNA-seq are relatively robust and unbiased techniques, I request that the authors provide whole-slide imaging data to allow for a more rigorous evaluation.

4. Regarding the authors' response to Point 7, I was asking the "Isotype control" but not "without primary antibody".

5. Regarding the authors' response to Point 8, according to the data provided by the authors: (1) HSPA5 was highly expressed in cluster 8 and cluster 10, rather than in the APCs as the authors claimed. (2) The expression levels of HSPA5 in macrophages were comparable to that in APCs, but after HHT treatment, both HSPA5 and SenMayo scores significantly increased in macrophages. This is inconsistent with the authors' claim that HHT eliminates HSPA5-expressing cells and reduces senescence markers. (3) If this senolytic effect is indeed cell type specific, then elucidating the molecular mechanism is critical for understanding their findings in vivo. (4) Although HSPA5 expression was reduced in APCs, SenMayo scores decreased in adipocytes, and the authors appear to conflate these two distinct observations.

In their response, the authors stated that HSPA5 expression positively correlates with SenMayo scores. I must reiterate that this claim is not supported by any data; specifically, the authors did not demonstrate that APCs with high SenMayo scores in the HF-PBS group expressed higher levels of HSPA5 compared to APCs with low SenMayo scores. The violin plots and bar graphs provided by the authors are insufficient to show their claim.

6. Regarding the authors' response to Point 9, based on the raw data provided by the authors, the absence of the siHSPA5 + Mock group makes it impossible to determine whether HSPA5 knockdown alone induces cell death. Moreover, the authors'

explanation does not clarify whether the survival of senescent cells is dependent on HSPA5 activity.

7. In Fig. 8G, the SA- β -Gal signal appears to localize to structures like bronchi or blood vessels, with nearly all cells within the circular region showing positive staining. Is the author attempting to suggest that all cells in this region are senescent? Furthermore, does the spatial distribution of p53 signal in this same area within the tissue?

Reviewer #5

(Remarks to the Author)

Version 1:

Reviewer comments:

Reviewer #1

(Remarks to the Author)

The authors have adequately and thoroughly addressed the comments raised, through addition of new data and edits. The data supporting HHT as a senotherapeutic and affecting WAT are strong.

Reviewer #2

(Remarks to the Author)

None

Reviewer #5

(Remarks to the Author)

RESPONSE to REVIEWER COMMENTS

Reviewer #1 (Remarks to the Author):

The authors have made a commendable effort in addressing the previous reviewer concerns and added substantial new data. I have a few more comments which, if addressed, would further improve the manuscript for publication.

Our response:

We sincerely thank the reviewer for the positive feedback and for recognizing our efforts to address the previous concerns with substantial new data. We also appreciate the additional constructive comments provided. We carefully addressed each of these points in the revised manuscript to further improve its clarity, rigor, and overall quality.

The study relies primarily on LDH and CCK-8 assays to test cytotoxicity of senescent cells. However, these methods are based on cellular metabolism, and senescent cells are known to have abnormal metabolic profiles that can skew such readouts. Please validate the cell viability/cytotoxicity data using metabolism-independent methods, such as nuclear staining for direct cell counting (DAPI/Hoechst) to corroborate findings.

Our response:

According to the reviewer's suggestion, non-senescent (NS) and replicative senescent (RS) HDFs and HPAs were incubated in culture media containing 0% FBS for 1 day and subsequently treated with the indicated concentrations of HHT for 4 days. After treatment, cells were stained with SA β G and eosin, and then cell numbers were quantified by directly counting cells from bright-field microscopic images using ImageJ (Figs. A–D, see below). This metabolism-independent approach yielded results consistent with those obtained using the CCK-8 assay (Figs. 1I and S3K). Furthermore, the concordance between CCK-8 assays and direct cell counting methods has been well established in previous studies (PMID: 31460316).

A HPAs under 0% FBS condition

B HDFs under 0% FBS condition

C HPAs under 0% FBS condition

D HDFs under 0% FBS condition

Common senescence markers of p16 and p21 are not interchangeable in many cases as they may represent distinct senescent subtypes and have different biological consequences. The authors used only p16 to evaluate senescence *in vitro* while they focused on p21 analysis *in vivo* tissues. It is unclear whether HHT targets both populations equally or selectively. It would be helpful to provide data on both p16 and p21 expression in the models studied.

Our response:

We thank the reviewer for raising this important point regarding the differential roles of p16 and p21 as senescence markers. In the revised manuscript, we included p21 data in Supplementary Figure 1 (Fig. S1) of our *in vitro* analyses, showing that the expression levels of p21 and p16 proteins were increased in senescent cells compared to non-senescent (NS) cells. For the human fat explant culture experiment shown in Figure 6N, we attempted to detect p16 by Western blotting but were unable to obtain reliable results, likely due to low protein abundance and technical limitations associated with this tissue type. Given these constraints, we focused on presenting p53 and p21 expression data for the *in vivo* human samples, as both are well-established markers of senescence and together provide robust evidence supporting our conclusions. In mouse samples, we have already included the levels of p53, p21, and p16 proteins. Although p16- and p21-positive cells may represent heterogeneous senescent subtypes, our data show that both markers were reduced by HHT treatment *in vitro* and *in vivo* mouse models. These findings suggest that HHT exerts senotherapeutic effects across both p16- and p21-defined populations.

Fig. S1 (B, D, F, and H, New data)

The results on macrophage M1/M2-like signatures appear inconsistent. While the authors provide some reasonable interpretations in the manuscript, another possibility is that a subset of macrophages may themselves become senescent, particularly in aged or metabolically stressed adipose tissue, and contribute to the confounding transcriptomic signals. It would be worthwhile to test whether senescent macrophages are present in these tissues and whether HHT treatment impacts this population.

Our Response:

We thank the reviewer for this insightful comment. We agree that a subset of macrophages may themselves become senescent, particularly in aged or metabolically stressed adipose tissue, and that this could contribute to the transcriptomic patterns observed in our study. While our current study was not designed to directly identify senescent macrophages, the increases in SenMayo and HSPA5 scores in macrophages following HHT treatment suggest that HHT's effects on this population may differ from its impact on adipose cell lineages (Cluster 0, 1, 3, please see violin plots and bar graphs below). These findings support the concept of cell type-specific

responses to HHT, in which macrophages may either undergo distinct senescence pathways themselves, or experience heightened functional stress following the clearance of senescent adipose cells. The latter scenario suggests that, after the removal of senescent or damaged adipose cells by HHT, surviving macrophages face a "greater burden of debris clearance", leading to increased functional stress and accelerated acquisition of senescent phenotypes (PMID: 38572110, 28721272).

In the revised manuscript, we expanded the Discussion and Limitations sections to include this interpretation, emphasizing that recognizing and understanding such cell type-specific effects is critical for evaluating the broader applicability and limitations of HHT as a therapeutic agent. We also consider the identification and characterization of senescent macrophages, as well as the assessment of their responses to HHT, to be an important focus for future studies.

These new sentences were added in the Discussion section.

“Consistently, we found that HHT treatment reduced the number of adipose macrophages, as evidenced by a decrease in crown-like structure (CLS). However, despite the significant reduction in total macrophage numbers, HHT-treated macrophages exhibited higher levels of age-associated gene signature (e.g., SenMayo) and a stronger M1-like signature, which contrasted with the overall reduction in tissue inflammation. This discrepancy may partly reflect methodological differences between transcriptomic and protein-based analyses. Recent evidence indicates that macrophage activation states can persist even after adipose inflammation resolves, due to epigenetic imprinting or stress adaptation^{79,80}. Accordingly, we propose that following HHT-induced apoptosis of senescent adipose cells, macrophages undergo transient stress responses associated with debris clearance, leading to pro-inflammatory and senescence-like features^{81,82}. These findings suggest that the observed signatures represent temporary adaptive states rather than persistent inflammation, highlighting the need for future studies to define cell type-specific responses to HHT. We further speculate that these discrepancies may arise from differences in macrophage status or disease conditions; for instance, in the published study⁸³, HHT suppressed intestinal inflammatory signatures when administered after inflammation had already been established, whereas in our study, mice were treated with HHT during the HF feeding period.”

Under 0% FBS conditions, cells may enter quiescence. Please check quiescent markers (e.g., p27KIP1, low Ki-67) to see if the control cells are simply quiescent as it would confound the interpretation of “selective killing”.

Our Response:

In accordance with the reviewer’s suggestion, we examined the expression of the proliferation marker Ki-67 in HPAs and HDFs cultured under 10% and 0% FBS conditions by Western blotting and immunofluorescence staining (Fig. A and B). Under 0% FBS, Ki-67 expression was very low in HPAs and undetectable in HDFs, indicating that the cells were in a quiescent state under these conditions. We acknowledged the potential quiescence of non-senescent cells, which may enable them to evade HHT-induced cell death.

We added the following sentences in the Limitation section of the revised manuscript.

"HHT-induced cytotoxicity was not observed in NS cells, particularly under serum-free (0% FBS) conditions. This resistance may be attributed to serum starvation-induced quiescence, which could enable cells to evade HHT's cytotoxic effects."

The strongest in vivo effects of HHT are seen in adipose tissue. Is this because HHT accumulates preferentially in fat, or that its target HSPA5 is expressed in adipose to a greater extent than other tissues? Biodistribution or PK/PD data would be informative.

Our response:

A previous study demonstrated that HSPA5 (GPR78) is highly expressed in adipose tissue (especially visceral adipose tissue) and is elevated with aging, obesity and diabetes in both humans and mice (please see below, adapted from PMID: 34615619). These findings support why adipose tissue is a major target organ of HHT.

According to a previous clinical study (PMID: 23053254), the mean Tmax on the first day of treatment was 0.55 hours, and the mean Cmax was 25.1 ng/mL. However, PK/PD data for HHT in mice are currently unavailable.

Aforementioned study (PMID: 34615619) identified HSPA5 as a binding partner of the SARS-CoV-2 spike protein, facilitating its interaction with angiotensin-converting enzyme 2 (ACE2) and thereby increasing the risk of severe COVID-19 progression. Consistent with this, recent studies have shown that HHT can limit SARS-CoV-2 progression (PMID: 34676091, 34162861), potentially through its targeting of HSPA5.

Together, these findings highlight HHT's potential to exert broad beneficial effects across metabolic, infectious, and age-associated pathologies, with adipose tissue as a particularly responsive site of action.

In summary, we are only at the early stages of understanding how HHT improves metabolic health and mitigates age-associated disease. Further mechanistic investigations, in vivo biodistribution, and PK/PD analyses are warranted to fully delineate its tissue-specific activity and translational potential.

We added the following sentences in the Limitation section of the revised manuscript.

“We observed the most pronounced in vivo effects of HHT in adipose tissue. A plausible explanation is that HSPA5 is highly expressed in adipose tissue, particularly in visceral adipose tissue, and its expression increases with aging, obesity, and diabetes in both humans and mice⁹⁰. These findings support the notion that adipose tissue may serve as a major target organ of HHT. Another possibility is that HHT preferentially accumulates in adipose tissue compared to other organs. However, tissue distribution and pharmacological action of HHT in mice are currently unavailable. Therefore, future pharmacokinetic and pharmacodynamic studies are warranted to determine the tissue distribution of HHT and to clarify its primary target sites.”

Figures from PMID: 34615619

To further support *HSPA5* as the senotherapeutic target of HHT, the authors should provide data showing *HSPA5* expression in senescent vs non-senescent cells, and across the major tissues affected. This would better support the proposed mechanism of action.

Our response:

In response to the reviewer's comment, we conducted additional experiments to further characterize *HSPA5* expression. We found that *HSPA5* protein levels were higher in senescent cells than in non-senescent (NS) cells across HPAs, HDFs, and hRPEs (Fig. S11E, please see below). Moreover, we examined *HSPA5* expression in other major metabolic tissues, including skeletal muscle (tibialis anterior) and liver, from chow-fed and high-fat diet (HF)-fed mice. *HSPA5* protein levels were significantly increased in both adipose tissue and skeletal muscle of HF-fed mice compared with chow-fed controls, whereas its expression remained largely unchanged in the liver (Fig. 7N and S11J). The more pronounced increase in *HSPA5* expression within adipose tissue suggests that adipose tissue is a primary metabolic target organ responsive to HHT treatment.

Fig. S11E (New data)

Fig. S11J (New data)

We added the following sentence in the Result section of the revised manuscript.

"HSPA5 protein levels were consistently higher in senescent cells than in NS cells across HPAs, HDFs, and hRPEs (Fig. S11E)."

"In metabolic tissues, HSPA5 protein levels were significantly increased in both adipose tissue and skeletal muscle from HF-induced obese mice compared with chow-fed controls, whereas its expression was not significantly changed in the liver (Figs. 7N and S11J). The increase in HSPA5 expression was more prominent in adipose tissue than in skeletal muscle, suggesting that adipose tissue is a primary metabolic target organ responsive to HHT treatment."

To further investigate whether HSPA5 serves as a senotherapeutic target of HHT, both NS and RS HPAs were treated with 100 nM HHT for 2 days—a time window selected to capture the induction and progression phase of apoptosis without inducing overt cell death—and subsequently subjected to co-immunostaining for HSPA5 and cleaved caspase-3 (Casp-3). HSPA5 was detected using a horse radish peroxidase (HRP)-conjugated anti-mouse secondary antibody with 3,3'-diaminobenzidine (DAB) chromogen, whereas Casp-3 was visualized using an alkaline phosphatase (AP)-conjugated anti-rabbit secondary antibody with Fast Red chromogen. Consistently, RS HPAs displayed higher HSPA5 and Casp-3 levels than NS HPAs (Fig. 7G). Notably, among the HHT-treated RS HPAs, cells with high HSPA5 expression exhibited a significantly greater number of Casp-3-positive cells compared to those with low HSPA5 expression (Fig. 7H). Furthermore, HHT treatment in HF-fed obese mice led to reduced

HSPA5 protein levels in adipose tissues compared with PBS-treated controls, likely due to the selective elimination of HSPA5 high-expressing senescent cells, as confirmed by immunohistochemistry (Fig. 7O). Collectively, these additional data strengthen our mechanistic insight and support the conclusion that HSPA5 might act as a key mediator of the senotherapeutic activity of HHT in HPAs.

Fig. 7G, 7H, and 7O (New data)

We added the following sentences in the Result section of the revised manuscript.

“To further confirm whether the senotherapeutic activity of HHT might be mediated through HSPA5, NS and RS HPAs were treated with 100 nM HHT for 2 days—a time window selected to capture the induction and progression of apoptosis without causing overt cell death—and subsequently subjected to co-immunostaining for HSPA5 and Casp-3, using DAB (brown) and Fast Red (red) as chromogens, respectively. RS HPAs treated with HHT consistently showed higher expression of HSPA5 and Casp-3 than HHT-treated NS HPAs (Fig. 7G). Among HHT-treated RS HPAs, cells with high HSPA5 expression showed a significantly greater number of Casp-3-positive cells compared with those expressing low HSPA5 (Fig. 7H). These data suggest that HSPA5 high expressing cells are primary target of HHT, and HSPA5 may serve as a key mediator of the senotherapeutic activity of HHT.”

HHT extends lifespan in Zmpste24^{-/-} and aged mice - median lifespan, maximum lifespan, or both? And how much % increase?

Our response:

Although Figures 8A, 9B, and 9G already present these data, in accordance with the reviewer's suggestion, we additionally calculated the median and maximum lifespans. In both male and female *Zmpste24^{-/-}* mice (Fig. 8A), HHT treatment extended the median lifespan from 14 to 15 weeks and the maximum lifespan from 15 to 19 weeks, corresponding to a 7.14% increase in median lifespan and a 26.67% increase in maximum lifespan. In aged mice (Fig. 9B), the median lifespan of 24 months could be determined only in the PBS-treated group. This is because, after 50% of the PBS-treated mice had died and a statistically significant difference in survival was observed between the PBS and HHT groups, the remaining animals were sacrificed for blood, urine, and tissue analyses rather than being followed until natural death. In an additional cohort (Fig. 9G), the median lifespan was 509 days in the PBS group and 551 days in the HHT group, corresponding to an 8.25% increase following HHT treatment. However, the maximum lifespan could not be determined because the mice were sacrificed at 558 days for blood, urine, and tissue analyses.

We included the followings in the Results section of the revised manuscript.

“HHT treatment significantly extended lifespan, increasing the median lifespan from 14 to 15 weeks and the maximum lifespan from 15 to 19 weeks (Fig. 8A).”

“We found that HHT significantly extended the lifespan of mice, increasing the median lifespan from 509 to 551 days, whereas the maximum lifespan could not be determined because the mice were sacrificed at 558 days for tissue collection (Fig. 9G).”

Minor point. Please define abbreviations (Ep, RP, SC, and BAT) in Figure 2C and the main text where they first appear. And please provide the PCR primer sequences used in the method section.

Our response:

We defined the above-mentioned abbreviations in the Figure legend of Fig. 2. We have already included the PCR primer sequences in Supplementary Table 9.

Reviewer #2 (Remarks to the Author):

In the revised manuscript “Homoharringtonine, a novel senotherapeutic, mitigates diet- and age-associated obesity and insulin resistance and extends the lifespan of mice”, Kim et al have indeed addressed the insufficient description of their experimental methods. However, in terms of experimental content, they still fail to convince me that they truly understand the molecular mechanism of how HHT eliminates senescent cells. Although the effects of drug treatment seem to be clear in animal experiments, the data and arguments are inconsistent and logically inadequate when authors investigate the impact of HHT on the immune microenvironment. These inconsistencies may suggest either instability in data quality or that the authors’ hypothesis regarding the senolytic effect of HHT is not entirely valid, thereby diminishing the clinical relevance of the study. However, if the focus is solely on HHT’s effects in a high-fat diet model, I believe the authors have made a considerable effort in this aspect.

Our response:

We thank the reviewer for this constructive and thoughtful comment. As correctly noted, our primary focus in this study is to investigate the effects of HHT in the high-fat diet (HFD) model, where we observed robust improvements in adipose tissue inflammation, insulin resistance, and overall metabolic health. Although we have also observed similar effects in human adipose tissue *ex vivo*, we fully agree that the clinical relevance of HHT requires future validation in well-designed translational studies.

We acknowledge the reviewer’s concern regarding certain inconsistencies in our macrophage data. In this revised version, we have carefully reanalyzed these datasets and provided detailed explanations of the possible biological reasons underlying the observed variability (e.g., differences in tissue context, treatment duration, or cell-type-specific responses).

We believe these clarifications now offer a more coherent interpretation of how HHT modulates immune (i.e., macrophages) and senescent cell interactions. We sincerely appreciate the reviewer’s insightful feedback, which has significantly improved the clarity and depth of our manuscript. We hope that the revisions adequately address the reviewer’s concerns and convey a more complete understanding of HHT’s senotherapeutic actions in metabolism- and age-associated pathological conditions.

1. Regarding the authors’ response to Point 2, I now understand the reason for using 0% FBS. In both studies of ABT263 and D+Q, apoptosis is a critical indicator for evaluating senolytic efficacy, rather than changes in cell number. Therefore, in conclusion, I believe the authors are not required to perform experiments under 0% FBS conditions (for reasons to be explained below). However, they do need to clearly demonstrate whether the reduction in cell numbers for senescent and normal cells is due to cell cycle arrest or apoptosis. However, since 100 nM HHT reduced the number

of NS HPAs and HDFs by approximately 40% and 25%, respectively, it is essential for the authors to describe how HHT passed the “toxicity to NS cells” screening threshold in Fig. S1I.

Our response:

Under both 0% and 10% FBS conditions, the senolytic effects of HHT on RS HPAs and HDFs were mediated by apoptosis, as evidenced by increased cleaved caspase-3 and decreased PARP expression in Western blot analyses (Figs. 1K, S3D, S3H, and S3M). Consistently, LDH activity in the culture media was elevated in these cells (Figs. 1J, S3C, S3G, and S3L). In contrast, non-senescent cells showed no detectable cleaved caspase-3, no change in PARP expression, and no increase in LDH activity, indicating the absence of apoptosis (Figs. 1J, 1K, S3C, S3D, S3G, S3H, S3L, and S3M). Furthermore, flow cytometry demonstrated that HHT treatment induced cell cycle arrest in non-senescent cells under 10% FBS conditions (Fig. S3I). Collectively, these findings indicate that HHT selectively induces apoptosis in senescent cells, whereas it primarily causes cell cycle arrest in non-senescent cells.

We would like to clarify that HPAs were not included in the initial screening processes for toxicity and senolytic assays. In the case of HDFs, treatment with HHT (#34 in the Table below) showed $45.19 \pm 1.35\%$ cell survival in RS HDFs and $96.59 \pm 2.42\%$ in NS HDFs during the initial screening. Therefore, HHT passed the “toxicity to NS cells” threshold, which we defined as $\geq 75\%$ cell survival. Although the data in Fig. S3F (previously Fig. S1I) indicate that HHT treatment resulted in approximately a 25% reduction in cell number for NS HDFs, we considered this acceptable because it still meets our predefined criterion for toxicity. We believe that the apparent discrepancy in toxicity between the initial screening and the subsequent experiment is due to differences in the cellular state, which likely influence the sensitivity of NS HDFs to HHT treatment.

No	Cell survival (%)						SAβG (comparable to rapamycin)					
	HUVECs		HDFs		hRPEs		RS HUVECs		RS HDFs		PS hRPEs	
	RS	NS	RS	NS	PS	NS						
ABT263	45.11 ± 1.45	96.59 ± 8.29	98.90 ± 2.28	98.76 ± 1.27	99.25 ± 4.32	98.10 ± 1.16						
15	101.63 ± 1.99	108.05 ± 5.54	89.18 ± 1.60	99.26 ± 1.60	77.01 ± 3.26	99.84 ± 2.36	+	+	+	-	-	-
34	82.08 ± 1.95	95.40 ± 5.36	45.19 ± 1.35	96.59 ± 2.42	98.67 ± 2.18	100.10 ± 1.64	+	+	+	-	-	+
39	91.75 ± 1.85	87.10 ± 3.63	81.69 ± 0.64	79.11 ± 2.87	79.06 ± 7.19	97.86 ± 1.29	+	+	+	+	+	-
114	102.34 ± 1.77	105.25 ± 2.28	87.99 ± 0.45	100.17 ± 6.17	76.07 ± 5.13	99.89 ± 1.03	+	+	+	+	+	+
124	98.27 ± 1.19	96.74 ± 0.58	86.34 ± 3.98	100.12 ± 4.18	78.24 ± 3.67	94.99 ± 1.31	+	+	+	+	+	+
136	101.43 ± 2.62	100.53 ± 2.41	89.47 ± 1.83	96.46 ± 3.90	80.76 ± 2.71	97.95 ± 0.94	-	-	-	+	+	+
142	92.90 ± 1.12	102.93 ± 1.47	86.07 ± 3.28	88.33 ± 2.76	nt	nt	-	-	-	+	+	+
151	99.08 ± 1.72	112.03 ± 1.55	103.45 ± 2.65	118.27 ± 1.72	93.17 ± 1.70	98.25 ± 2.20	-	-	-	-	-	+
158	100.51 ± 2.18	117.02 ± 1.70	95.59 ± 1.59	106.33 ± 1.33	80.47 ± 0.51	100.78 ± 0.67	+	+	+	+	+	+
160	96.84 ± 1.60	105.98 ± 2.16	89.24 ± 1.08	97.21 ± 1.22	71.41 ± 1.21	97.62 ± 1.44				+	+	-
162	97.66 ± 1.93	117.29 ± 1.79	82.19 ± 2.23	98.59 ± 2.69	69.97 ± 1.61	96.33 ± 1.97	+	+	+	+	+	+
165	95.01 ± 0.44	111.77 ± 2.52	87.28 ± 2.73	106.78 ± 6.04	74.08 ± 0.98	94.17 ± 0.14	+	+	+	+	+	-
167	57.64 ± 3.57	91.90 ± 8.64	54.63 ± 1.74	88.29 ± 1.62	55.10 ± 0.71	86.76 ± 1.20	-	-	-	-	-	-
168	97.35 ± 1.68	104.72 ± 2.45	87.14 ± 0.79	108.41 ± 5.95	79.79 ± 2.01	97.60 ± 0.65	+	+	+	+	+	-
169	100.81 ± 1.38	108.44 ± 2.73	85.20 ± 2.07	101.00 ± 1.05	79.56 ± 1.19	97.09 ± 0.28	+	+	+	+	+	-
170	101.02 ± 1.97	113.70 ± 1.71	86.64 ± 1.95	102.50 ± 3.01	75.84 ± 4.05	94.90 ± 4.02	+	+	+	+	+	-
171	96.54 ± 1.68	123.87 ± 4.47	84.38 ± 2.05	87.77 ± 1.85	76.13 ± 5.54	99.92 ± 2.85	+	+	+	+	+	-
199	20.55 ± 2.56	106.39 ± 9.09	31.54 ± 8.89	17.38 ± 1.44	24.90 ± 0.35	19.52 ± 0.81	-	-	-	-	-	-
213	1.60 ± 0.58	2.00 ± 0.99	41.80 ± 4.19	89.67 ± 1.18	17.58 ± 1.43	20.93 ± 0.82	-	-	-	-	-	-
215	70.60 ± 3.24	87.20 ± 1.98	77.28 ± 5.63	89.53 ± 5.88	61.78 ± 2.58	84.97 ± 4.45	+	+	+	+	+	+
219	115.95 ± 2.07	124.23 ± 8.20	94.91 ± 0.74	110.40 ± 6.23	85.04 ± 7.08	96.62 ± 0.95	-	-	-	+	+	-
223	65.34 ± 2.28	99.75 ± 2.32	80.17 ± 3.35	109.22 ± 11.36	65.95 ± 2.92	84.57 ± 1.27	-	-	-	+	+	+
231	60.43 ± 1.85	97.36 ± 1.55	77.92 ± 0.85	90.68 ± 7.49	61.92 ± 4.87	87.71 ± 1.31	-	-	-	+	+	+
327	81.46 ± 4.43	103.29 ± 3.26	99.29 ± 2.90	98.18 ± 5.27	80.80 ± 2.18	91.06 ± 5.82	+	+	+	+	+	-
351	93.19 ± 3.10	102.24 ± 1.81	97.51 ± 2.66	99.42 ± 5.45	100.00 ± 11.27	103.49 ± 2.25	+	+	+	+	+	-
352	91.24 ± 5.09	110.66 ± 5.71	54.92 ± 3.62	95.07 ± 1.38	90.65 ± 6.13	99.66 ± 2.26	+	+	+	-	-	-
376	91.75 ± 1.25	91.62 ± 7.38	96.26 ± 2.80	95.17 ± 5.21	88.22 ± 3.92	101.84 ± 3.08	-	-	-	+	+	-
386	100.11 ± 2.99	87.08 ± 2.73	79.60 ± 2.34	96.79 ± 2.69	51.32 ± 1.61	89.92 ± 1.88	+	+	+	+	+	-
399	87.95 ± 1.74	97.39 ± 3.48	79.67 ± 2.12	95.09 ± 1.44	87.09 ± 0.56	98.94 ± 1.14	-	-	-	+	+	-
405	91.01 ± 3.93	93.69 ± 3.00	81.59 ± 1.33	91.84 ± 1.75	85.48 ± 4.22	88.60 ± 1.30	-	-	-	+	+	-
407	94.71 ± 0.33	92.67 ± 5.22	80.53 ± 2.24	110.03 ± 1.22	86.48 ± 1.11	93.00 ± 2.25	-	-	-	+	+	-
419	89.96 ± 3.13	95.41 ± 1.74	80.28 ± 1.57	86.62 ± 1.38	82.24 ± 1.69	94.74 ± 0.31	-	-	-	+	+	-
441	33.83 ± 1.89	91.24 ± 1.90	78.33 ± 1.08	91.96 ± 2.46	94.22 ± 4.23	96.97 ± 2.38	-	-	-	+	+	+
442	94.82 ± 1.30	99.43 ± 2.78	84.85 ± 1.59	88.55 ± 2.60	92.08 ± 1.18	88.12 ± 0.75	-	-	-	+	+	-
466	95.03 ± 0.40	96.24 ± 4.67	89.97 ± 1.09	85.55 ± 1.25	102.28 ± 1.27	88.32 ± 0.38	-	-	-	+	+	+
496	41.65 ± 0.60	90.36 ± 1.21	51.94 ± 0.56	31.24 ± 0.37	84.27 ± 2.35	97.09 ± 2.10	-	-	-	-	-	-
499	98.94 ± 1.11	103.95 ± 3.20	76.55 ± 2.13	89.24 ± 1.39	89.73 ± 3.54	95.11 ± 1.44	-	-	-	+	+	-
501	93.02 ± 1.52	107.88 ± 5.05	71.93 ± 2.94	82.15 ± 1.86	81.78 ± 4.09	91.10 ± 2.76	-	-	-	+	+	-
502	96.51 ± 2.06	106.63 ± 2.14	78.42 ± 1.43	90.51 ± 1.45	89.73 ± 0.98	96.34 ± 1.13	-	-	-	+	+	-
503	97.04 ± 0.57	92.28 ± 4.92	78.83 ± 2.91	90.69 ± 0.78	86.91 ± 3.80	95.74 ± 1.21	-	-	-	+	+	-
510	2.43 ± 1.19	3.01 ± 1.51	32.76 ± 1.31	86.81 ± 0.83	21.22 ± 1.10	15.05 ± 0.87	-	-	-	-	-	-
529	91.23 ± 2.54	106.57 ± 5.08	88.57 ± 3.89	100.18 ± 4.18	99.86 ± 5.80	99.43 ± 1.61	-	-	-	+	+	-
533	95.14 ± 2.20	106.25 ± 4.96	86.82 ± 2.67	86.95 ± 1.31	90.44 ± 6.27	91.00 ± 1.05	-	-	-	+	+	-
584	34.36 ± 1.08	101.46 ± 0.83	90.40 ± 2.17	88.18 ± 1.88	102.78 ± 5.31	86.39 ± 2.97	-	-	-	+	+	+
600	57.23 ± 1.96	91.81 ± 2.02	94.61 ± 3.83	102.94 ± 1.76	99.13 ± 6.47	96.18 ± 3.04	-	-	-	-	-	-
692	59.53 ± 5.47	92.19 ± 3.09	76.94 ± 2.22	82.51 ± 2.32	56.12 ± 8.48	98.83 ± 3.37	-	-	-	-	-	-
726	97.27 ± 6.00	102.65 ± 3.73	89.72 ± 4.30	97.02 ± 1.23	96.25 ± 1.92	104.02 ± 4.21	-	-	-	+	+	-
740	100.75 ± 5.70	98.29 ± 5.01	88.80 ± 3.17	98.49 ± 1.03	97.24 ± 3.92	93.80 ± 1.76	-	-	-	+	+	-
743	106.69 ± 7.30	99.53 ± 2.23	91.69 ± 1.40	86.64 ± 2.81	103.56 ± 6.09	93.77 ± 2.52	-	-	-	+	+	-
753	99.58 ± 4.27	99.00 ± 2.71	84.48 ± 1.23	98.10 ± 1.33	86.35 ± 6.66	94.64 ± 3.28	-	-	-	+	+	-
765	101.45 ± 5.22	107.60 ± 6.01	90.14 ± 2.67	96.50 ± 2.06	93.59 ± 5.75	90.28 ± 2.64	-	-	-	+	+	-
772	78.93 ± 7.65	110.62 ± 9.21	96.36 ± 5.20	94.23 ± 2.41	91.15 ± 8.20	96.23 ± 3.37	-	-	-	-	-	+
837	107.54 ± 7.48	101.35 ± 3.28	95.89 ± 3.32	97.56 ± 2.26	106.94 ± 9.13	91.18 ± 2.08	-	-	-	+	+	+

HHT. This quiescent state may enable cells to evade HHT's cytotoxic effects through reduced metabolic activity and cell cycle arrest. In fact, the Reviewer 1 also asked a similar question and suggested to test quiescent markers (e.g., p27KIP1 or low Ki-67). Under 0% FBS, Ki-67 expression was very low in HPAs and undetectable in HDFs, indicating that the cells were in a quiescent state under these conditions (Figure A and B, please see below). Additionally, we cannot exclude the possibility that autophagy activation under serum-depleted conditions contributed to drug resistance, as autophagy is known to serve as a cytoprotective mechanism against various cellular stresses, including chemotherapeutic agents. We acknowledged the potential quiescence and autophagy activation of non-senescent cells, which may enable them to evade HHT-induced cell death, as a limitation of our study.

We included the following sentence in the Limitation section of the revised manuscript.

"HHT-induced cytotoxicity was not observed in non-senescent (NS) cells, particularly under serum-free (0% FBS) conditions. This resistance may be attributed to serum starvation-induced quiescence, which could enable cells to evade HHT's cytotoxic effects. Additionally, autophagy activation under serum-depleted conditions may have contributed to drug resistance, as autophagy functions as a well-established cytoprotective mechanism against various cellular stresses, including chemotherapeutic agents."

We appreciate the reviewer's insightful comment regarding the potential influence of serum deprivation on HSPA5 expression and translational activity. We acknowledge that low-serum conditions may alter HSPA5 levels or induce autophagy-related adaptive responses, which could in theory contribute to differential drug sensitivity. However, these aspects were beyond the primary scope of our current study because the vast majority of our experiments, including those assessing HSPA5 expression, protein synthesis, and cellular responses to HHT, were conducted under standard culture

conditions with 10% FBS, where cells maintain normal proliferative and translational profiles. The serum-free condition (0% FBS) was employed only as a transient quiescent control to demonstrate that HHT is not cytotoxic to non-senescent cells arrested in the cell cycle. Thus, our key conclusions are derived from physiologically relevant serum conditions rather than from serum-deprived states. We fully agree that future studies should examine how serum deprivation or autophagy induction may modulate HSPA5 expression and contribute to drug resistance mechanisms. Such investigations will further clarify whether autophagy-mediated adaptations influence HHT sensitivity in different cellular contexts.

2. As the authors mentioned, if HHT reduces p53 and p21 levels by inhibiting translation, this mechanism does not explain how HHT induces G1 cell cycle arrest in normal cells. Furthermore, it also fails to illustrate why p53 and p21 levels remain unchanged in 0% FBS-treated NS cells, as shown in Fig. 1K. In addition, the description of Fig. 1K was missing in the manuscript.

Our response:

In the current study, we did not investigate the specific mechanism underlying HHT-induced G1 arrest in normal cells. Given that HHT is a well-established inhibitor of protein translation in actively proliferating cells (PMID: 30975912), it is plausible that it reduces the synthesis of key cell-cycle regulators, thereby contributing to G1 arrest. Regarding the unchanged p53 and p21 levels in 0% FBS-treated NS cells (Fig. 1K), our temporal analysis showed that both proteins decreased during the first day of HHT treatment and gradually returned to baseline by day 4. This transient change likely reflects an adaptive or compensatory response of NS cells to sustained translation inhibition, potentially via feedback mechanisms that restore cell-cycle checkpoint control. We included these new temporal data in our rebuttal.

We have already mentioned about Fig. 1K in the main text and the Figure legend of the original manuscript as follows:

“Under these conditions, HHT induced significant cytotoxicity in senescent HPAs and HDFs while having no effect on NS cell numbers (Figs. 1H-K and S3J-M).” – manuscript page 6

“K. Expression levels of age- or apoptosis-associated proteins (n=3 in each group).” – manuscript page 54

3. Regarding the authors' response to Point 6, they cited several references to argue that changes in the M1/M2 ratio within adipose tissue may not reliably reflect the inflammatory state. Through this reasoning, they attempt to rationalize the inconsistency between the increased M1/M2 ratio observed in the snRNA-seq data and the results from histological analyses. However, among the cited literature, one study explicitly states that although classical markers may not clearly distinguish M1 and M2 macrophages, the MacSpectrum characterization can still identify M1-like and M2-like phenotypes (PMID: 30990466). This study also confirmed that in obese mice, M1-like macrophages increase while M2-like macrophages decrease.

If the authors insist that neither classical markers nor transcriptomic signatures can indicate the inflammatory status of macrophages in adipose tissue, then why do they present a separate analysis of M1 and M2 populations in the CyTOF data shown in Fig. S15B, claiming a significant decrease in M1 macrophages? Is it possible that the authors separated M1 and M2 populations in the analysis because combining them would not have resulted in statistical significance? In addition, the authors stated that “to precisely assess the inflammatory status of ATMs, they analyzed CLS using F4/80 staining.” However, as the authors mentioned, F4/80 is a pan-macrophage marker expressed by various macrophage subtypes. In fact, if expression level is to be considered, pro-inflammatory macrophages—such as M1 or certain monocyte-derived populations—may express even lower levels of F4/80 compared to M2 or tissue-resident macrophages, as noted in the references cited by the authors and others (PMID: 37651193). Furthermore, according to the CyTOF result, M1 macrophages constituted only a small fraction of the total macrophage population, which was strongly inconsistent with the F4/80 staining patterns observed in CLS. Because both CyTOF and snRNA-seq are relatively robust and unbiased techniques, I request that the authors provide whole-slide imaging data to allow for a more rigorous evaluation.

Our response:

We thank the reviewer for these detailed comments and the opportunity to clarify this point. We believe there may have been a misunderstanding regarding our analyses: we did not use the “ratio” of M1- to M2-like macrophages in any of our data or in our previous response. We fully acknowledge that computational characterization using MacSpectrum can distinguish M1-like and M2-like phenotypes. Indeed, this was the reference (PMID: 30990466) we relied upon for our macrophage analyses (Fig. S8D) and in our response to Point 6.

We wish to emphasize that the decision to present M1 and M2 populations separately in Fig. S15B was not driven by statistical considerations. Rather, this analysis was

performed in response to one of the reviewer's earlier suggestions and aligns with our overall strategy of employing multiple complementary approaches—including Western blotting of NF- κ B pathway components, cytokine profiling, CyTOF, and immunohistochemical staining—to comprehensively assess inflammatory status.

Regarding the reviewer's comment on F4/80 staining, we have now included immunofluorescence images using the M1-specific marker iNOS, which more directly reflects pro-inflammatory macrophages, instead of the F4/80-positive crown-like structures (Fig. S8E).

We revised the following sentences in the Result section of the revised manuscript.

“Despite the seemingly contradictory transcriptomic signatures, our immunofluorescent staining with the M1-like macrophage marker iNOS further confirmed that CLS with iNOS-positive pro-inflammatory macrophages was significantly reduced in HHT-treated WAT (Fig. S8E).”

Fig. S8E (New data)

E

To further strengthen our response and address the reviewer's request for greater rigor, we have also provided whole-slide imaging data of F4/80 staining in this rebuttal (please see below). We believe that these data, together with our multi-assay approach, offer a more comprehensive and reliable evaluation of macrophage-associated inflammation in HHT-treated adipose tissue (please see below).

We recognize that the reviewer's fundamental concern lies in the apparent inconsistency between M1/M2-like signatures and the systemic metabolic improvements with reduced adipose inflammation observed in HHT-treated mice. This discrepancy may in part reflect methodological differences between protein-based and transcriptomic approaches.

In addition, recent studies have shown that obesity-induced macrophage activation signatures may persist even after macrophage infiltration and adipose inflammation are resolved, likely due to epigenetic imprinting of adipose immune cells (PMID: 40634602, 35618862).

Our interpretation is that, following the removal of senescent or damaged adipose cells by HHT, adipose macrophages may experience an increased debris clearance burden, resulting in functional stress and accelerated acquisition of pro-inflammatory and senescence-like features (PMID: 38572110, 28721272), which may persist even after adipose inflammation are resolved,

Collectively, we speculate that the increased pro-inflammatory and senescence-associated transcriptome signatures observed in HHT-treated macrophages reflect persistent inflammatory programs initiated during prior HHT-induced adipose cell death (PMID: 40634602). These sustained programs may underlie the heterogeneity and apparent contradictions in our transcriptomic dataset.

In the revised manuscript, we have added new sentences in the Discussion section to include this interpretation and highlight the need for future studies to determine cell type-specific effects of HHT, particularly in macrophages.

“Consistently, we found that HHT treatment reduced the number of adipose macrophages, as evidenced by a decrease in crown-like structure (CLS). However, despite the significant reduction in total macrophage numbers, HHT-treated macrophages exhibited higher levels of age-associated gene signature (e.g., SenMayo) and a stronger M1-like signature, which contrasted with the overall reduction in tissue inflammation. This discrepancy may partly reflect methodological differences between transcriptomic and protein-based analyses. Recent evidence indicates that macrophage activation states can persist even after adipose inflammation resolves, due to epigenetic imprinting or stress adaptation^{79,80}. Accordingly, we propose that following HHT-induced apoptosis of senescent adipose cells, macrophages undergo transient stress responses associated with debris clearance, leading to pro-inflammatory and senescence-like features^{81,82}. These findings suggest that the observed signatures represent temporary adaptive states rather than persistent inflammation, highlighting the need for future studies to define cell type-specific responses to HHT. We further speculate that these discrepancies may arise from differences in macrophage status or disease conditions; for instance, in the published study⁸³, HHT suppressed intestinal inflammatory signatures when administered after inflammation had already been established, whereas in our study, mice were treated with HHT during the HF feeding period.”

4. Regarding the authors' response to Point 7, I was asking the "Isotype control" but not "without primary antibody".

Our response:

We now provide the isotype control data for the reviewer.

5. Regarding the authors' response to Point 8, according to the data provided by the authors: (1) HSPA5 was highly expressed in cluster 8 and cluster 10, rather than in the APCs as the authors claimed. (2) The expression levels of HSPA5 in macrophages were comparable to that in APCs, but after HHT treatment, both HSPA5 and SenMayo scores significantly increased in macrophages. This is inconsistent with the authors' claim that HHT eliminates HSPA5-expressing cells and reduces senescence markers. (3) If this senolytic effect is indeed cell type specific, then elucidating the molecular mechanism is critical for understanding their findings in vivo. (4) Although HSPA5 expression was reduced in APCs, SenMayo scores decreased in adipocytes, and the authors appear to conflate these two distinct observations.

Our responses:

We appreciate the reviewer's careful attention to these points and the opportunity to clarify our interpretations.

Regarding comment (1).

We acknowledge that clusters 8 and 10 express higher levels of HSPA5 compared to other cell types. In our dataset, cluster 8 shows higher expression of both SenMayo and HSPA5 than the APC population (cluster 2). Recent studies have identified mesothelial-like cells (e.g., cluster 8) within rodent visceral adipose tissue (such as perigonadal fat). These cells exhibit properties similar to APCs and have been implicated in adipose tissue fibrosis (PMID: 37886999, 36889280, 26412153). We believe that HHT may act on mesothelial-like cells in a manner similar to its effects on APCs, as we observed a trend toward reduced SenMayo and HSPA5 expression in cluster 8 following HHT treatment. Given that these mesothelial-like cells and cluster 10 remain poorly characterized minor populations, we chose to focus primarily on the well-defined APC population (cluster 2) in the main discussion, but we do not exclude the possibility that HHT also impacts other cell types, including clusters 8 and 10.

Regarding comments (2) & (3):

Our data suggest that HHT exerts a senolytic effect on certain adipose cell lineages, including APCs and adipocytes, but its effect on macrophages appears distinct. Specifically, HHT-treated macrophages displayed increased HSPA5 and SenMayo scores, suggesting that HHT does not have senolytic activity in this cell type and may even enhance senescence-like features. One plausible explanation is that, after the removal of senescent or damaged adipose cells, surviving macrophages experience a greater “debris clearance” burden *during the early phase of HHT treatment*, leading to increased functional stress and accelerated acquisition of senescence phenotypes (PMID: 38572110, 28721272).

HHT is known to act through multiple mechanisms, including inhibition of translation via binding to the A-site cleft of the 60S ribosomal subunit (PMID: 24501394), depletion of short-lived survival or oncogenic proteins such as MCL-1 (PMID: 20971952), induction of cell-cycle arrest at various phases (PMID: 33716735, 33548212), and inhibition of STAT3 signaling (PMID: 26166037). HHT has also been reported to alleviate macrophage-mediated inflammation through NF- κ B pathway inhibition (PMID: 36211988). However, direct evidence for HHT-mediated promotion of M1 macrophage activation or increased inflammatory signaling is limited. Our findings suggest that HHT may exert cell-type-specific effects, with macrophage responses potentially shaped more by microenvironmental demands than by direct drug action. Future studies are warranted to elucidate these differential mechanisms.

As already mentioned earlier, we incorporate this interpretation into the revised Discussion section as follows to highlight the possibility of cell type-specific responses to HHT or HHT-treated microenvironmental influence on macrophages.

“Consistently, we found that HHT treatment reduced the number of adipose macrophages, as evidenced by a decrease in crown-like structure (CLS). However, despite the significant reduction in total macrophage numbers, HHT-treated macrophages exhibited higher levels of age-associated gene signature (e.g., SenMayo) and a stronger M1-like signature, which contrasted with the overall reduction in tissue inflammation. This discrepancy may partly reflect methodological differences between transcriptomic and protein-based analyses. Recent evidence indicates that macrophage activation states can persist even after adipose inflammation resolves, due to epigenetic imprinting or stress adaptation^{79,80}. Accordingly, we propose that following HHT-induced apoptosis of senescent adipose cells, macrophages undergo transient stress responses associated with debris clearance, leading to pro-inflammatory and senescence-like features^{81,82}. These findings suggest that the observed signatures represent temporary adaptive states rather than persistent inflammation, highlighting the need for future studies to define cell type-specific responses to HHT. We further speculate that these discrepancies may arise from differences in macrophage status or disease conditions; for instance, in the published study⁸³, HHT suppressed intestinal inflammatory signatures when administered after inflammation had already been established, whereas in our study, mice were treated with HHT during the HF feeding period.”

Regarding comments (4).

We appreciate the opportunity to clarify this point. As the reviewer pointed out, although HSPA5 expression was reduced in APCs, SenMayo scores were decreased in adipocytes. It has been shown that young and healthy APCs under high-fat diet conditions can differentiate into mature adipocytes (PMID: 34562641, 22596050). In line with this, we observed an increased number of adipocytes in the HHT-treated group. Based on these findings, we speculate that healthy, young APCs with lower HSPA5 expression may have differentiated into new mature adipocytes, leading to an increase in newly formed adipocytes (Fig. 4A, 4B). This process, known as hyperplastic adipogenesis, represents a form of healthy adipose tissue remodeling (PMID: 31573549, 37181756) and may contribute to the observed reduction in SenMayo scores in adipocytes.

Our interpretation of cellular aging and inflammation does not rely on a single parameter or transcriptomic signature. Instead, we integrated multiple complementary assays—including protein-level analyses (NF- κ B, inflammatory cytokines), CyTOF, and histological evaluation, which consistently demonstrated reduced inflammatory signaling, decreased M1 macrophage abundance, and reduced senescence markers in adipose tissue following HHT treatment. While some transcriptomic signatures (e.g., SenMayo in macrophages) diverged from these protein-based observations, such differences are not unexpected and may reflect distinct biological contexts, assay sensitivities, or temporal dynamics.

In their response, the authors stated that HSPA5 expression positively correlates with SenMayo scores. I must reiterate that this claim is not supported by any data; specifically, the authors did not demonstrate that APCs with high SenMayo scores in the HF-PBS group expressed higher levels of HSPA5 compared to APCs with low SenMayo scores. The violin plots and bar graphs provided by the authors are insufficient to show their claim.

Our response:

Our statement that “Hspa5 expression positively correlates with SenMayo scores” was based on the observation that cell clusters with higher average SenMayo scores (e.g., Clusters 2, 8, and 10, please see below) also displayed higher average Hspa5 expression. This pattern was evident in our violin plots and bar graphs and was intended to highlight a cluster-level trend, rather than a cell-by-cell correlation within a given cluster. We agree with the reviewer that our current data do not directly demonstrate that, within the APC population of the HF-PBS group, cells with high SenMayo scores express higher Hspa5 levels than those with low scores. Nonetheless, we believe this cluster-level association reflects a biologically meaningful relationship between ER stress and cellular senescence, which we plan to examine in greater detail through future single-cell correlation analyses.

6. Regarding the authors' response to Point 9, based on the raw data provided by the authors, the absence of the siHSPA5 + Mock group makes it impossible to determine whether HSPA5 knockdown alone induces cell death. Moreover, the authors' explanation does not clarify whether the survival of senescent cells is dependent on HSPA5 activity.

Our responses:

We agree that including the siHSPA5+Mock group is important to properly assess the effectiveness of HHT in the absence of HSPA5. As shown in Figure A, no significant difference in cell number was observed between the scrambled siRNA (siSc)+Mock and siHSPA5+Mock groups. For this reason, we used the siSc+Mock group as the control in the experiments presented in Figs. 7C–E. We hope this clarification addresses the reviewer's concern regarding our interpretation of HHT's mechanism of action. If necessary, we can note this point in the Limitations section of the revised manuscript.

Regarding whether the survival of senescent cells depends on HSPA5 activity, previous studies have demonstrated that complete knockout of Grp78 (Hspa5) in mice causes embryonic lethality due to apoptosis of the inner cell mass (PMID: 16847323), and conditional knockout in the hematopoietic compartment results in cell death and impaired oncogenic signaling (PMID: 21937694). These findings establish HSPA5 as an essential endoplasmic-reticulum chaperone required for cell survival, whose complete disruption leads to severe cellular defects. Notably, partial depletion of HSPA5 by siRNA

or shRNA alters cellular plasticity and increases susceptibility to stress-induced cell death (PMID: 21937694, 39472623, 28130223). Collectively, these studies indicate that loss of HSPA5 can trigger cell death, although the outcome depends on cell type, knockdown efficiency, degree of senescence, and cellular stress context.

Moreover, to clarify whether the survival of senescent cells depends on HSPA5 activity, we performed additional experiment. In this new experiment, HPAs were transfected with higher concentration of siHSPA5 (25 nM). After 4 days of incubation, cells were co-treated with 500 nM doxorubicin (Dox) to enhance senescence induction and 100 nM HHT or PBS. Under this condition, the viability of siHSPA5+Mock cells was reduced to $81.07 \pm 2.54\%$ relative to siSc+Mock cells (Figure B), indicating that survival of senescent cells partially depends on HSPA5. Importantly, cell viability markedly decreased to $43.61 \pm 1.44\%$ in siSc+HHT cells but remained high ($81.94 \pm 2.15\%$) in siHSPA5+HHT cells, suggesting that a certain level of HSPA5 expression is required for the senolytic activity of HHT. Notably, there was no significant difference in cell viability between siHSPA5+Mock and siHSPA5+HHT groups. Collectively, these findings demonstrate that the senotherapeutic effect of HHT is strongly dependent on the presence of HSPA5.

To further investigate whether HSPA5 serves as a senotherapeutic target of HHT, both NS and RS HPAs were treated with 100 nM HHT for 2 days—a time window selected to capture the induction and progression phase of apoptosis without inducing overt cell death—and subsequently subjected to co-immunostaining for HSPA5 and cleaved caspase-3 (Casp-3). HSPA5 was detected using a horse radish peroxidase (HRP)-conjugated anti-mouse secondary antibody with 3,3'-diaminobenzidine (DAB) chromogen, whereas Casp-3 was visualized using an alkaline phosphatase (AP)-conjugated anti-rabbit secondary antibody with Fast Red chromogen.

Consistently, RS HPAs displayed higher HSPA5 and Casp-3 levels than NS HPAs (Fig. 7G). Notably, among the HHT-treated RS HPAs, cells with high HSPA5 expression exhibited a significantly greater number of Casp-3-positive cells compared to those with low HSPA5 expression (Fig. 7H). Furthermore, HHT treatment in HF-fed obese mice led to reduced HSPA5 protein levels in adipose tissues compared with PBS-treated controls, likely due to the selective elimination of HSPA5 high-expressing senescent

cells, as confirmed by immunohistochemistry (Fig. 7O). Collectively, these additional data strengthen our mechanistic insight and support the conclusion that HSPA5 might act as a key mediator of the senotherapeutic activity of HHT in HPAs.

Fig. 7G, 7H, and 7O (New data)

We added the following sentences in the Result section of the revised manuscript.

“To further confirm whether the senotherapeutic activity of HHT might be mediated through HSPA5, NS and RS HPAs were treated with 100 nM HHT for 2 days—a time window selected to capture the induction and progression of apoptosis without causing overt cell death—and subsequently subjected to co-immunostaining for HSPA5 and Casp-3, using DAB (brown) and Fast Red (red) as chromogens, respectively. RS HPAs treated with HHT consistently showed higher expression of HSPA5 and Casp-3 than HHT-treated NS HPAs (Fig. 7G). Among HHT-treated RS HPAs, cells with high HSPA5 expression showed a significantly greater number of Casp-3-positive cells compared with those expressing low HSPA5 (Fig. 7H). These data suggest that HSPA5 high expressing cells are primary target of HHT, and HSPA5 may serve as a key mediator of the senotherapeutic activity of HHT.”

We also included the following sentence in the Discussion section of the revised manuscript

“Our findings suggest that the senotherapeutic effect of HHT is highly dependent on HSPA5, as its knockdown markedly attenuated the pro-apoptotic activity of HHT. This indicates that HHT requires HSPA5 as a functional target or signaling mediator to induce cell death, and that the physical interaction between HHT and HSPA5 may be critical for initiating apoptosis. In contrast, simple depletion of HSPA5 expression by siRNA may allow compensatory chaperone networks or

stress-adaptive responses to preserve cell survival. Moreover, siRNA reduces protein levels gradually and incompletely, whereas HHT may acutely disrupt specific HSPA5 functions, including those at the endoplasmic reticulum or cell surface, that are essential for maintaining proteostasis and stress signaling. Thus, the loss of HHT-induced cytotoxicity under HSPA5 knockdown conditions supports the conclusion that HSPA5 is not only associated with but also required as a functional binding partner for HHT-mediated apoptosis, highlighting a mechanistic dependency rather than a redundant pathway.”

7. In Fig. 8G, the SA- β -Gal signal appears to localize to structures like bronchi or blood vessels, with nearly all cells within the circular region showing positive staining. Is the author attempting to suggest that all cells in this region are senescent? Furthermore, does the spatial distribution of p53 signal in this same area within the tissue?

Our responses:

We appreciate the reviewer’s observation and the opportunity to clarify this point. We do not intend to imply that all cells within the SA- β -Gal–positive region are senescent. SA- β -Gal staining has known limitations, including background activity in metabolically active or lysosome-rich cells (PMID: 17634571, 26105537, 20010931). As recommended in the literature (PMID: 20010931, 39121846), senescent cell identification is more reliable when supported by multiple, complementary markers. For this reason, we employed additional assays, including p53 immunostaining, Western blotting for p53, p21, and p16, and trichrome staining. SA- β -Gal staining was performed on frozen tissue sections, whereas p53 immunostaining was conducted on paraffin-embedded sections from the same cohort of mice. Because these assays were performed on non-identical sections, direct spatial correlation between SA- β -Gal and p53 signals is not possible. Nonetheless, both markers indicate the presence of senescent cells in the tissue, with expected differences in their distribution due to both methodological factors (different sample preparation) and biological variability between markers.

A point-by-point response to the reviewers' comments

Reviewer #1 (Remarks to the Author):

The authors have adequately and thoroughly addressed the comments raised, through addition of new data and edits. The data supporting HHT as a senotherapeutic and affecting WAT are strong.

Response: We thank Reviewer #1 for the positive comment. We are pleased that our revisions and newly added data have adequately addressed the concerns raised, and we believe the reviewer's comments have substantially strengthened the manuscript.

Reviewer #2 (Remarks to the Author):

None

Response: We thank Reviewer #2 for their time and consideration of our revised manuscript.

Reviewer #5 (Remarks to the Author):

Response: We thank Reviewer #5 for their time and contribution to the peer-review process.

We also sincerely appreciate the Nature Communications editorial team for arranging and facilitating the peer-review process, as well as for their thorough editorial handling of our manuscript throughout the submission and revision process.